# Modelling of thrombus formation using smoothed particle hydrodynamics method

**Alessandra Monteleone[1]☉, Alessia Viola[1,2]☉, Enrico Napoli[2], Gaetano Burriesci ![ORCID][1,3]***

**1** Ri.MED Foundation, Palermo, Italy, **2** Engineering Department, University of Palermo, Palermo, Italy,
**3** UCL Mechanical Engineering, University College London, London, United Kingdom

☉ These authors contributed equally to this work.
* g.burriesci@ucl.ac.uk

**Data Availability Statement:** All relevant data are within the paper.

**Funding:** The funders had no role in study design, data collection and analysis, decision to publish, or preparation of the manuscript.

## Abstract

In this paper a novel model, based on the smoothed particle hydrodynamics (SPH) method, is proposed to simulate thrombus formation. This describes the main phases of the coagulative cascade through the balance of four biochemical species and three type of platelets. SPH particles can switch from fluid to solid phase when specific biochemical and physical conditions are satisfied. The interaction between blood and the forming blood clot is easily handled by an innovative monolithic FSI approach. Fluid-solid coupling is modelled by introducing elastic binds between solid particles, without requiring detention and management of the interface between the two media. The proposed model is able to realistically reproduce the thromboembolic process, as confirmed by the comparison of numerical results with experimental data available in the literature.

## 1. Introduction

Cardiovascular diseases (CVDs) are the main causes of mortality in the world, representing 30% of all global deaths [1]. Hence, there is a pressing need to develop novel tools to diagnose and manage these dysfunctions. Thrombosis is one of the main causes of CVD, that results in the formation of a clot that can obstruct the physiological blood circulation, or fragment and flow through the cardiovascular system to critical organs, thus causing disorders such as ictus, stroke or pulmonary embolism.

Thrombus formation and its pathological role have been investigated for centuries [2]; however, only in the second half of the nineteenth century some fundamental progress was made to provide some systematic understanding of the process. In 1856 Virchow [3] published his observations on the influence of blood flow conditions on platelet activation and, consequently, on thrombus formation. This led to the elaboration of the *Virchow's triad*, that identifies changes in blood components (*hypercoagulability*), vessel wall surface (*endothelial injury*) or flow characteristics (*stasis*) as synergic contributors to the phenomenon. Nowadays this theory is still accepted and used to predict the process.

The coagulation process is a sequence of events designed to limit potential blood losses, thus leading to hemostasis. Hemostasis is a complex physiological process due to the interaction between blood, platelets, clotting factors and coagulation inhibitors. The process can be subdivided into primary and secondary hemostasis. In the first phase, adhesion, activation,

**Competing interests:** NO authors have competing interests.

and aggregation of platelets occur [4]. The result is a biological structure called platelet plug. Platelets, that represent the key parameters leading to thrombosis, can be activated by a long exposure to high shear stress or chemical agonist enzymes like adenosine diphosphate or thromboxane. Another platelet activation mechanism is the interaction with the Von Willebrand factor affecting the adhesion of platelets to the injured tissue walls [5].

The coagulative cascade is the process behind the thrombus formation and represents the heart of secondary hemostasis. The clotting enzymes, called factors, are present in the blood in the inactive form and indicated with the roman numerals XII, XI, IX, VIII, VII, X, V, II, I. They can be switched to the active form (XIIa, XIa, IXa, VIIIa, VIIa, Xa, Va, IIa, Ia) through complex biochemical interactions. In particular, the clotting process can be triggered via intrinsic or extrinsic pathways, both leading to fibrin activation. Specifically, factor XIIa and factor VIIa/Tissue Factor (TF) are triggering variables of the intrinsic and the extrinsic pathways, respectively, and both start the common path of factor Xa/Va. Each of these enzymes activates the formation of the complex prothrombin (II) which catalyses the conversion into thrombin (IIa). The latter acts as an enzyme by transforming fibrinogen (I) into filaments of fibrin (Ia) that trap platelets, blood cells and plasma, causing the formation of a fibrin clot, that gets deposited on the platelet plug mesh (previously activated from primary hemostasis).

Thrombus is a clot that forms inside a vessel when the hemostatic process becomes abnormally activated, and can result in a partial or complete obstruction of the vessel lumen. When the lumen is only partial barred, the altered hemodynamics caused by the narrowing may cause the growth of the clot and, eventually, levels of shear stress and pressure differences sufficient to detach the full clot of parts of it from the anchor point. These masses, released in the bloodstream, can travel to smaller vessels and obstruct the blood supply to the downstream tissues, causing their death.

More commonly, clots associated with primary hemostasis generate in the arterial system around injured atherosclerotic plaques and usually occur at regions of high shear flow. They are characterised by predominance of platelets (hence also referred to as white thrombi), and are common cause of myocardial infarction and stroke. Clots associate with secondary hemostasis may arise without endothelial wall damage and are promoted by areas of slow flow shear rate typically occurring in the venous system. These are characterised by predominance of red cells (hence also called red thrombi), and are a common cause of venous thromboembolism and pulmonary embolism [6].

Given the complexity of the process and the large number of chemical or physical variables involved, numerical modelling represents an attractive tool to simulate the phenomenon of thrombosis. Microscopic and macroscopic thrombus models, analysing the mechanism at different scales, have been proposed in the literature [7]. Zhang *et al.* [8] and Gao *et al.* [9] implemented a novel multiscale approach based on discrete particle methods to model thrombus formation in cardiovascular diseases by coupling the macroscopic flow conditions with cellular and molecular effects of platelet mechanical activation. Xu *et al.* [10] proposed a multi-scale approach where fluid was simulated on the macro-scale using dissipative particle dynamics, and the fine-scale receptors' biochemical reactions were modelled by coarse-grained molecular dynamics. Most of the macroscopic models are based on Computational Fluid dynamics (CFD) analysis, where convection-diffusion equations are solved to describe the interaction between blood flow and chemical or biological agents involved in thrombosis process [11]. In this framework, Sorensen *et al.* [12] defined a novel model able to show the mechanism of platelet activation and aggregation in the proximity of to the vessel wall, including the action of chemical agonists species, through a weight function. Moreover, they emphasised the key role of thrombin during thrombus formation. Leiderman and Fogelson [13] presented a novel continuum blood clotting model to reproduce the interactions among the main chemical

species, the platelet concentration and flow transport aspects linked to shear rate. Anand *et al.* [14] described the formation and dissolution of blood clot using biochemical reactions and rheological factors, focusing on the role of fibrin.

CFD simulations have also been used to understand complex pathologies involving thrombus formation. In this context, Sarrami *et al.* [4] developed a computational model to predict the thrombogenic dynamics in intracranial aneurysms treated with flow-diverter devices. Menichini *et al.* [15] proposed a novel hemodynamics-based model to predict the formation of thrombus in type B aortic dissection, where shear rate, fluid residence time and platelet distribution were used to evaluate thrombosis and simulate its growth. Vella *et al.* [16] evaluated thromboembolic risk in left atrial appendage under atrial fibrillation pathology using an ideal geometry and imposing the wall movement under different blood flow conditions.

Although in the literature there are numerous examples of mathematical models or CFD techniques that describe blood clotting or thrombus formation, fluid-structure interaction (FSI) approaches have recently been applied to describe the process more realistically. Generally, FSI is used to model multiphysics phenomena that occur in systems where the fluid flow and deformable structure interact synergically, as in common cardiovascular applications [17]. FSI approaches can be classified into partitioned and monolithic. In partitioned FSI, two different solvers are used to describe the fluid and solid phases, and the mutual interaction occurs at the interface separating the two domains. In particular, fluid flows are commonly described using Eulerian formulations, whilst solid are modelled through Lagrangian approaches. The coupling of the fluid and solid domains is commonly achieved employing Arbitrary-Lagrangian-Eulerian (ALE) techniques [18–21] or Immersed Boundary (IB) strategies [22]. On the other hand, in monolithic methods both solid and fluid domains are treated with a unique solver and no interface is required. In this case, Fully Eulerian [23, 24] or Lagrangian [25–29] formulations are adopted in the whole domain.

Recently, a number of particle techniques had been developed to describe thrombosis. Tsubota *et al.* [30] presented a semi-implicit two-dimensional moving particle approach to model thrombus formation after Fontan surgery. In this model, fluid particles are converted into solid phase by adding internal spring forces when blood stasis condition occurs (the model does not consider biochemical factors). Masalceva *et al.* [31] developed a two-dimensional particle-based model including thrombus shell as aggregate of particles and thrombin specie. Also this approach neglects the biochemical reactions of the coagulation cascade and fibrin formation. Wang *et al.* [32] proposed a novel particle method to simulate thrombus formation employing a velocity decay factor linked to the fibrin concentration, to take into account the interaction with blood. Wang *et al.* [33] developed a dissipative particle dynamics model to study the adhesion and aggregation process of injured platelets on the collagen surface, by incorporating a model of high non-physiological shear stresses traumatised platelets to a viscoelastic model.

In the context of the Lagrangian particle method, smoothed particle hydrodynamics (SPH) has recently been adopted for the modelling of thrombosis considering the influence of blood flow, platelets and biochemical factors [34, 35]. Chui *et al.* [36] used SPH to model the adhesion and the aggregation of clot particles and the mechanism governed by physical triggers (i.e. low shear stress). Al Saad *et al.* [37] idealised a SPH model to describe thrombus formation, where platelet adhesion and aggregation is obtained through elastic forces that depend exclusively on geometrical distances from an injured vessel. Ariane *et al.* [38] proposed a two-dimensional model to simulate the interaction between blood flow and emboli-like structures in a double venous valve system. In this approach no particle agglomeration is used to model emboli structures, that are considered as fixed. This technique was extended by Baksamawi *et al.* [39], including an algorithm based on geometrical distance for particle agglomeration.

This paper presents a new three-dimensional numerical method based on SPH for the simulation of thrombus formation. Contrary to previous works, the proposed approach efficiently combines the biomechanical and biochemical processes in the thrombosis phenomenon. Four biochemical species and three platelets states are considered to replicate the main phases of the coagulative cascade. A particle agglomeration/dissolution algorithm is proposed, able to model both thrombus formation and growth, as well as embolisation. The fluid-solid coupling is enforced through to the inclusion of elastic forces between solid particles, which are established by recruiting fluid particles when specific hydrodynamic and biochemical conditions are satisfied. An innovative monolithic FSI approach is developed to describe the interaction between blood and the forming thrombus using a single solver. The model is implemented in the open-source package PANORMUS (PArallel Numerical Open-souRce Model for Unsteady flow Simulations) [40]. The proposed approach was validated comparing the results with experimental data available in literature [41].

## 2. SPH formulation

In the SPH method, the domain is described through a finite number N of particles having own mass, density, volume and other physical properties. The field variables at each particle are obtained using discrete convolution integrals with filter functions of assigned shape, named kernel functions W. The kernel function has a characteristic length named smoothing length, indicated as h, which controls the influence domain of W. Each i particle has a support domain, $\Omega_i$, which includes all the surrounding particles having distance from the position of i ($\mathbf{x}_i$) lower than the product between h and a scalar factor k, whose value depends on the shape of the specific kernel function. In this study, the Wendland function was used [42], where the proportionality constant k between the radius of the support domain and the smoothing length h is equal to 2. The total number of particles N depends on the isotropic starting distance $\Delta_x$, which is commonly assigned as proportional to the smoothing length h. In this study this distance was assumed equal to $\Delta_x = kh/2$ as reported by [40, 43, 44]. The hydrodynamic variable a computed at the position $\mathbf{x}_i$ of the i particle can be expressed as

$$a_i = \sum_{j=1}^{N_i} \frac{m_j}{\rho_j} f\left(\boldsymbol{x}_j\right) W_{ij} \tag{1}$$

where Ni is the number of j particles lying into $\Omega_i$; $m_j$ and $\rho_j$ are the mass and density of j; and $W_{ij} = W(\mathbf{x}_i - \mathbf{x}_j, h)$.

In SPH simulations of incompressible flows, the weakly compressible (WCSPH) and truly incompressible (ISPH) approaches can be used. In the WCSPH scheme, a thermodynamics equation of state is introduced to relate pressure and density. In the ISPH algorithm the pressure field is obtained implicitly by solving a system of Pressure Poisson Equations (PPEs), following the projection method proposed by Chorin [45], thereby satisfying the incompressibility severely. In this study, the ISPH scheme is employed, where a fractional-step procedure is used to solve the momentum and continuity equations. For a detailed description see [46]. In the first step of the procedure, named predictor-step, the momentum equation is solved removing the pressure gradient term, so as to obtain the intermediate velocity u*. In SPH approximation, this equation can be written as

$$\frac{\boldsymbol{u}_i^* - \boldsymbol{u}_i^{(r)}}{\Delta t} + \frac{3}{2} Diff_i^{(r)} - \frac{1}{2} Diff_i^{(r-1)} - \boldsymbol{f}_i = 0 \tag{2}$$

where $\Delta t$ is the time step, the index r indicates the time instant, $\mathbf{u}_i^*$ is the intermediate velocity

of the i particle, $\mathbf{u}_i^r$ is the velocity at time r, and $\mathbf{f}_i$ is the mass force per unit mass acting on the i particle. In Eq 2, $Diff_i$ is the diffusive term which is calculated using the Adams–Bashforth scheme [47] to obtain a second-order accurate explicit approximation

$$Diff_i = -\sum_{j=1}^{N_i} m_j \left(v_i + v_j\right) \frac{\left(\boldsymbol{x}_i - \boldsymbol{x}_j\right) \cdot \nabla W_{ij}}{d_{ij}^2} \left(\boldsymbol{u}_i - \boldsymbol{u}_j\right) \tag{3}$$

where $\nabla W_{ij}$ is the gradient of the kernel function and $d_{ij}$ is the distance between the i and j particles.

The PPE are then solved to obtain the corrective velocity

$$\sum_{j=1}^{N_i} \frac{2 \ m_j}{\rho_j} \frac{\left(\boldsymbol{x}_i - \boldsymbol{x}_j\right) \cdot \nabla W_{ij}}{d_{ij}^2} \left(\psi_i - \psi_j\right) = \frac{1}{\Delta t} \sum_{j=1}^{N_i} \frac{m_j}{\rho_j} \left(\boldsymbol{u}_i^* - \boldsymbol{u}_j^*\right) \cdot \nabla W_{ij} \tag{4}$$

where $\psi$ is the pseudo-pressure, which has the dimension of the kinematic pressure $\left(\text{P}/_{\varrho}\right)$.

The updated velocity $\mathbf{u}_i^{(r+1)}$ is finally obtained as

$$\boldsymbol{u}_i^{(r+1)} = \boldsymbol{u}_i^* - \Delta t \sum_{j=1}^{N_i} \frac{m_j}{\rho_j} \left(\psi_i - \psi_j\right) \nabla W_{ij} \tag{5}$$

The boundary at solid walls is treated adopting a mirror particles procedure, based on the mirroring of the particles in the vicinity of the wall, to impose suitable boundary conditions and overcome the truncation of the kernel function at the walls. A detailed description of the procedure is provided in [40]. The inflow/outflow boundaries are treated following the approach described in [46].

## 3. The proposed thrombus formation model

### 3.1 Biochemical species and platelets activation

The coagulative cascade takes a leading role in thrombus formation. As shown in Fig 1, a large number of factors are involved in process. In this study, four biochemical species were considered, to simplify and simulate the clot formation: prothrombin (pt), thrombin (th), fibrinogen (fg) and fibrin (fi).

These species, highlighted in red in the figure, come into play at the conclusive stage of the coagulative cascade, when the formed fibrin interacts with activated platelets. The proposed model also includes three platelet states: resting platelets (rp), activated platelets (ap) and fibrin bound aggregated platelets (bp).

The transport of the modelled species through the flow domain was evaluated by solving the convection-diffusion equation

$$\frac{\Delta C_s}{\Delta t} - \alpha_s \nabla^2 C_s - S_s = 0 \tag{6}$$

where C is the time dependent concentration of the generic specie (s = pt, th, fg, fi, rp, ap, bp), $\alpha_s$ and $S_s$ are the diffusivity and the source term, respectively, and $\Delta C/\Delta t$ is the total derivative operator that, in the Lagrangian formulation, includes the convective term. In SPH, the concentration is associated at each mass point. Therefore, for the generic i particle the value of the

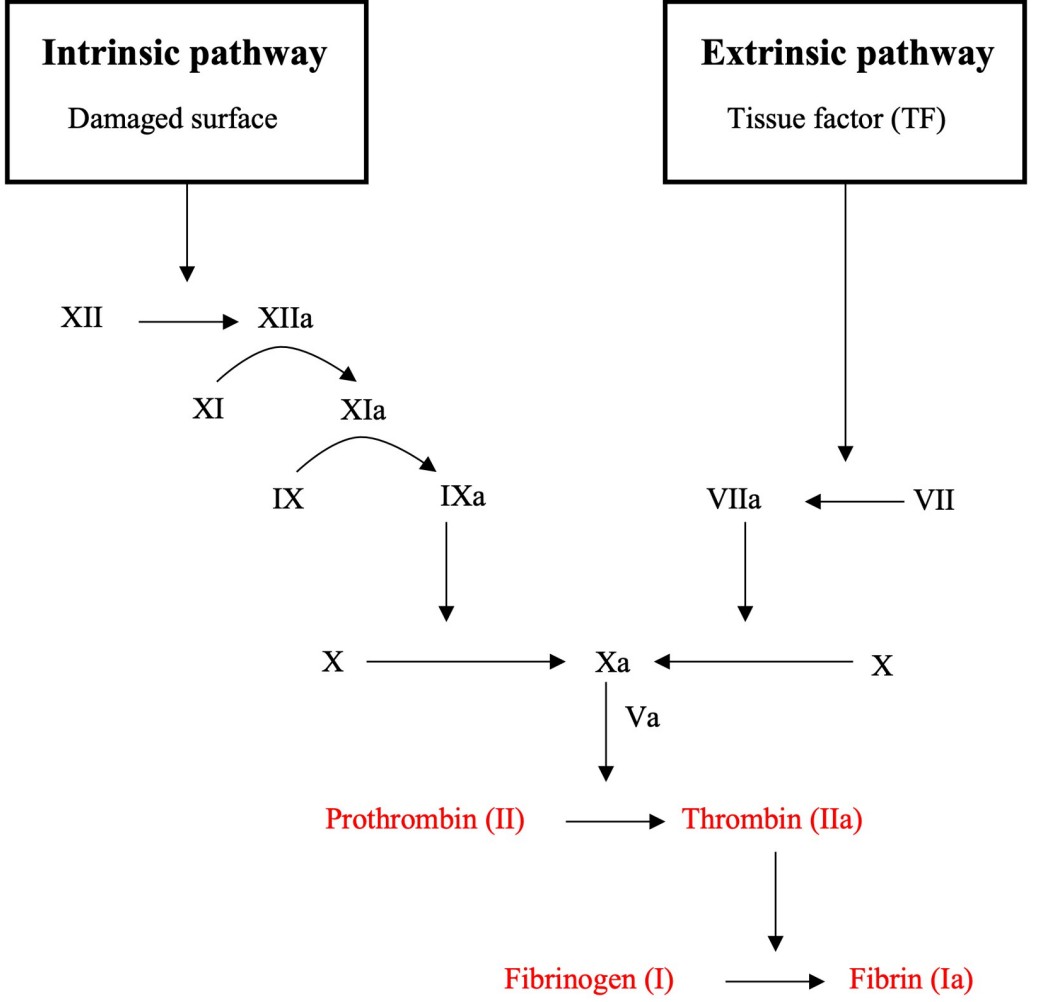

**Fig 1. Schematic view of the coagulative cascade.**

concentration for the species s, $C_{i,s}$, can be calculated at the updated time step (r+1) as

$$C_{i,s}^{(r+1)} - C_{i,s}^{(r)} - \alpha_s \left( \frac{3 Diff_i^{(r)}}{2} - \frac{Diff_i^{(r-1)}}{2} \right) \Delta t + S_s \, \Delta t = 0 \qquad (7)$$

with the diffusive term $Diff_i$

$$Diff_i = -\sum_{j=1}^{N_i} m_j \left( v_i + v_j \right) \frac{\left( \boldsymbol{x}_i - \boldsymbol{x}_j \right) \cdot \nabla W_{ij}}{d_{ij}^2} \left( C_{i,s} - C_{j,s} \right) \qquad (8)$$

where $C_{j,s}$ is the concentration of s associated to the j particle lying in the support domain of i and the other symbols are known.

The source terms for the modelled species are summerised below

$$S_{th} = k_{th}^{rp} C_{rp} C_{pt} + k_{th}^{ap} C_{ap} C_{pt} \tag{9}$$

$$S_{pt} = -S_{th} \tag{10}$$

$$S_{fi} = \frac{k_{fi}^{th} C_{th} C_{fg}}{K_{m,fi}^{th} + C_{fg}} \tag{11}$$

$$S_{fg} = -S_{fi} \tag{12}$$

$$S_{ap} = k_{ap} C_{rp} - k_{bp} \phi_{pb}^{fi} C_{ap} \tag{13}$$

$$S_{rp} = -k_{ap} C_{rp} \tag{14}$$

$$S_{bp} = k_{bp} \phi_{pb}^{fi} C_{ap} \tag{15}$$

The conversion of prothrombin to thrombin was assumed to occur on the surface of resting and activated platelets having kinetic constants $k_{th}^{rp}$ and $k_{th}^{ap}$, respectively (see Table 1). According to Rosing [48], activated platelets support thrombin generation at a much higher rate than resting platelets, thus different values of kinetic constants were adopted for rp and ap. Thrombin promotes the conversion of fibrinogen to fibrin and the activation of resting platelets. The generation of fibrin promoted by thrombin was assumed to follow Michaelis-Menten kinetics [49] with kinetic constants $k_{fi}^{th}$ and $K_{m,fi}^{th}$ (see Table 1). For the activation of platelets, a function of local agonist concentrations was employed for the kinetic constant $k_{ap}$, as discussed in [12].

**Table 1. Values of reactions kinetic, diffusive coefficients and initial concentrations adopted in the model.**

| Biochemical reactions kinetic constants | | | |
|---|---|---|---|
| **Symbol** | **Value** | **UM** | **References** |
| $k_{th}^{rp}$ | $6.50 \cdot 10^{-10}$ | U PLT$^{-1}$ s$^{-1}$ μM$^{-1}$ | [12] |
| $k_{th}^{ap}$ | $3.69 \cdot 10^{-10}$ | U PLT$^{-1}$ s$^{-1}$ μM$^{-1}$ | [12] |
| $K_{m,fi}^{th}$ | 3160 | nM | [53] |
| $k_{fi}^{th}$ | 59 | s$^{-1}$ | [53] |
| $k_{pb}$ | $1 \cdot 104$ | s$^{-1}$ | [13] |
| $C_{th}^*$ | $9.11 \cdot 10^{-10}$ | M | [12] |
| $C_{fi,50}$ | 600 | nM | [4] |
| **Diffusion Coefficients** | | | |
| $D_{th}$ | $6.47 \cdot 10^{-7}$ | cm$^2$ s$^{-1}$ | [53] |
| $D_{fi}$ | $2.47 \cdot 10^{-7}$ | cm$^2$ s$^{-1}$ | [53] |
| $D_{ap}$ | $2.50 \cdot 10^{-7}$ | cm$^2$ s$^{-1}$ | [13] |
| $D_{bp}$ | 0 | cm$^2$ s$^{-1}$ | [4] |
| **Biochemical Initial Concentrations** | | | |
| $C_{pt}$ | 1400 | nM | [4] |
| $C_{fg}$ | 7000 | nM | [4] |
| $C_{rp}$ | $2 \cdot 10^8$ | PLT ml$^{-1}$ | [4] |
| $C_{ap}$ | $1 \cdot 10^7$ | PLT ml$^{-1}$ | [4] |

This function evaluates the impact of each single specie on the rate of platelet activation and can be expressed as

$$k_{ap} = \begin{cases} 0, & \Omega < 1 \\ \dfrac{\Omega}{t_{act}}, & \Omega \geq 1 \end{cases} \tag{16}$$

where $t_{act}$ is the platelets activation time equal to 1 s [12]. The activation function $\Omega$ can be written as

$$\Omega = \sum_{s=1}^{N_s} w_s \frac{C_s}{C_s^*} \tag{17}$$

where $w_s$ is the weight assigned to the agonist s, $C_s$ is the concentration of the agonist s and $C_s^*$ is the threshold value for the concentration of s. In the proposed model, thrombin was considered as the only agonist specie, as it typically contributes to most of the platelet activation phenomenon. Hence, Eq 17 reduces to

$$\Omega = w_{th} \ \frac{C_{th}}{C_{th}^*} \tag{18}$$

where $w_{th} = 1$ and $C_{th}^*$ is the threshold concentration for the thrombin. The latter was set equal to $9.11 \cdot 10^{-10}$ M, as recommended in the literature [12, 50] (see Table 1). Therefore, conversion from resting to activated platelets is achieved when the concentration of the thrombin reaches the threshold value $C_{th}^*$.

Activated platelets aggregate to the fibrin network to form bound platelets and, thus, the clot.

Following [4], thrombin-induced fibrin generation and its effect on platelet trapping and aggregation is modelled by means of a second order Hill function $\phi_{pb}^{fi}$ that describes processes involving cooperative binding events [51].

$$\phi_{bp}^{fi} = \frac{C_{fi}^2}{C_{fi}^2 + C_{fi,50}^2} \tag{19}$$

where $C_{fi,50}$ is the half-saturation constant (concentration of fibrin where the half-maximal occupation occurs) [51].

The constants and parameters used in the model are obtained from experimental studies available in the literature [4, 12, 48, 52–54]. These values, listed in Table 1, are general and applicable in different flow conditions [4, 12].

## 3.2 Trigger factor and boundary conditions

A trigger factor can be used as the start condition for thrombus formation. Since thrombin is considered to have the greater contribution to the coagulation process (leading to platelet activation, fibrin mesh generation and bound platelets formation), the triggering is imposed as flux boundary conditions for thrombin concentration at the injured wall.

Fig 2 summarises, with a 2D sketch, the boundary conditions imposed for the modelled species at inflow/outflow boundaries and at healthy and injured walls.

Specifically, at the healthy vessel walls (including inflow/outflow boundaries), homogeneous Neumann conditions are imposed for the concentration of all the species $\frac{\partial C_s}{\partial n} = 0$, where n is the direction normal to the boundary (pointing towards the interior of the domain). This is achieved by imposing the concentrations of the mirror particle equal to that of the

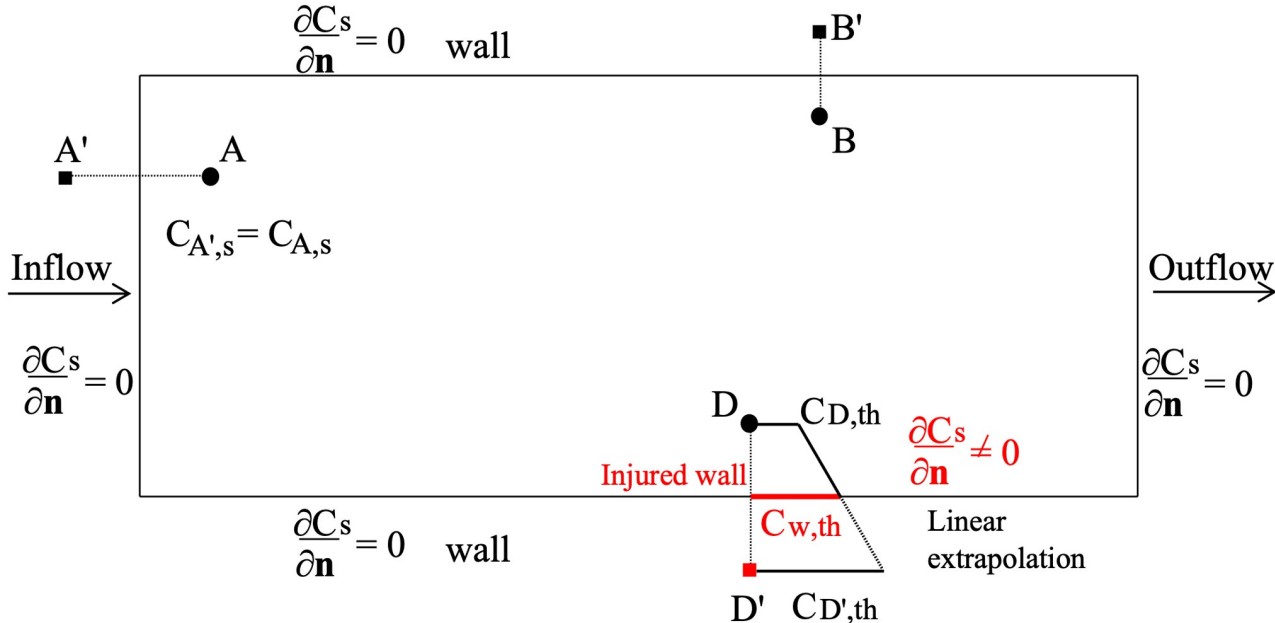

**Fig 2. Boundary conditions for the species involved in the coagulation cascade process.** Circles: effective particle; squares: mirror particles; red bold line: injured wall.

generating particle. In Fig 2 the concentration of mirror particle B' (black square), generated through wall boundary, is equal to the concentration of the generating particle B (black circle). The same occurs for inflow and outflow boundaries (mirror particles A' and E', respectively). On the other hand, at the injured wall region (red bold line in Fig 2), a non-null value for the normal derivative of thrombin concentration is imposed. To this aim, the concentration of the D' mirror particle (red square) generated through the injured wall is obtained through a linear extrapolation based on the value of the generating particle D and the assigned concentration of thrombin at the injured wall, indicated as $C_{w,th}$

$$C_{D',th} = 2C_{w,th} - C_{D,th} \tag{20}$$

### 3.3 Thrombus formation

In the proposed model, the concentrations of pt, th, fg, fi, rp, ap and bp are evaluated at each particle through the convection-diffusion described by Eq 7, with the source terms in Eqs 9–15. Moreover, it was assumed that the concentrations of the species do not affect the blood velocity in normal conditions, in the absence of thrombus. On the other hand, a monolithic fluid-structure interaction approach is used when fluid blood particles convert to solid particles, mimicking the presence of thrombus.

The scheme represented in Fig 3 describes the proposed thrombus formation model. Considering the generic i particle, the concentration of thrombin $C_{i,th}$ is calculated solving the convection-diffusion Eq 7 with source term from Eq 9. Eq 7 has very low kinetic reaction in the bulk, whilst it is accelerated at the injured wall introducing a flux boundary condition for thrombin, as described in section 3.2. Thrombin promotes the conversion of fibrinogen $C_{i,fg}$ into fibrin $C_{i,fi}$ through the source term of Eq 11. Thrombin allows the conversion of resting platelets into activated platelets (Eq 7 + source term in Eq 13). Since the function $\Omega$ is greater than 1 (Eq 18) and the kinetic constant $k_{pa}$ in Eq 16 is, therefore, greater than 0, the activation of platelets begins when the concentration of thrombin $C_{i,th}$ exceeds the threshold value $C_{th}^*$.

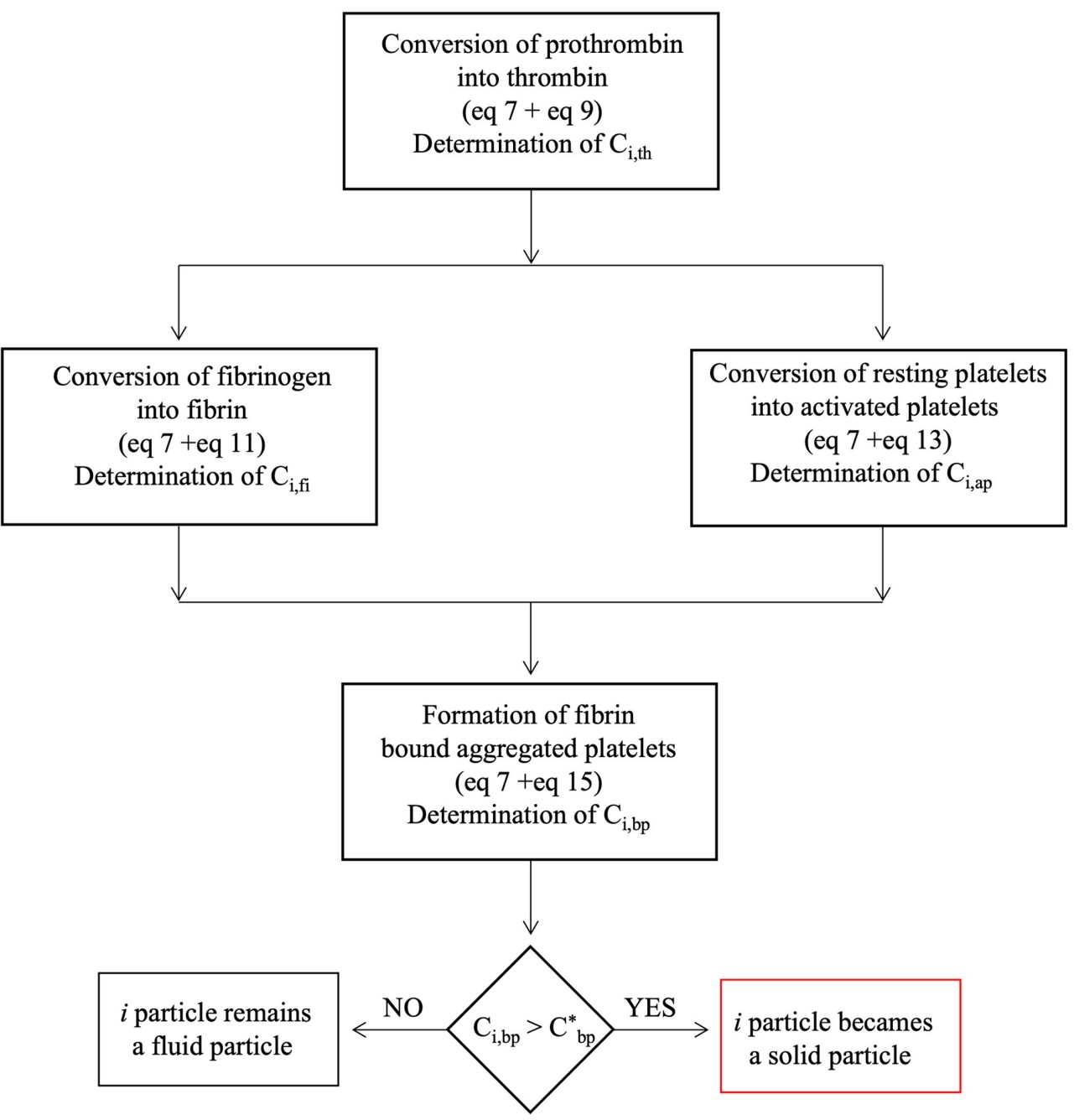

**Fig 3. Schematic representation of the proposed thrombus model.**

Activated platelets connected to the fibrin network generate bound platelets (Eq 7 + source term of Eq 15). Therefore, if the concentration of bound platelet $C_{i,bp}$ exceeds a set limit $C_{bp}^*$, then the i fluid particle is switched to a solid particle by enforcing spring connections with the neighboring solid particles. On the other hand, if $C_{i,bp}$ is below $C_{bp}^*$, i remains a fluid particle.

In Fig 4a, which represents a snapshot at time instant r, all the represented particles have a concentration of bound platelets lower than the threshold value $C_{bp}^*$. These particles (represented as full black circles) are treated as fluid particle following the ISPH formulation

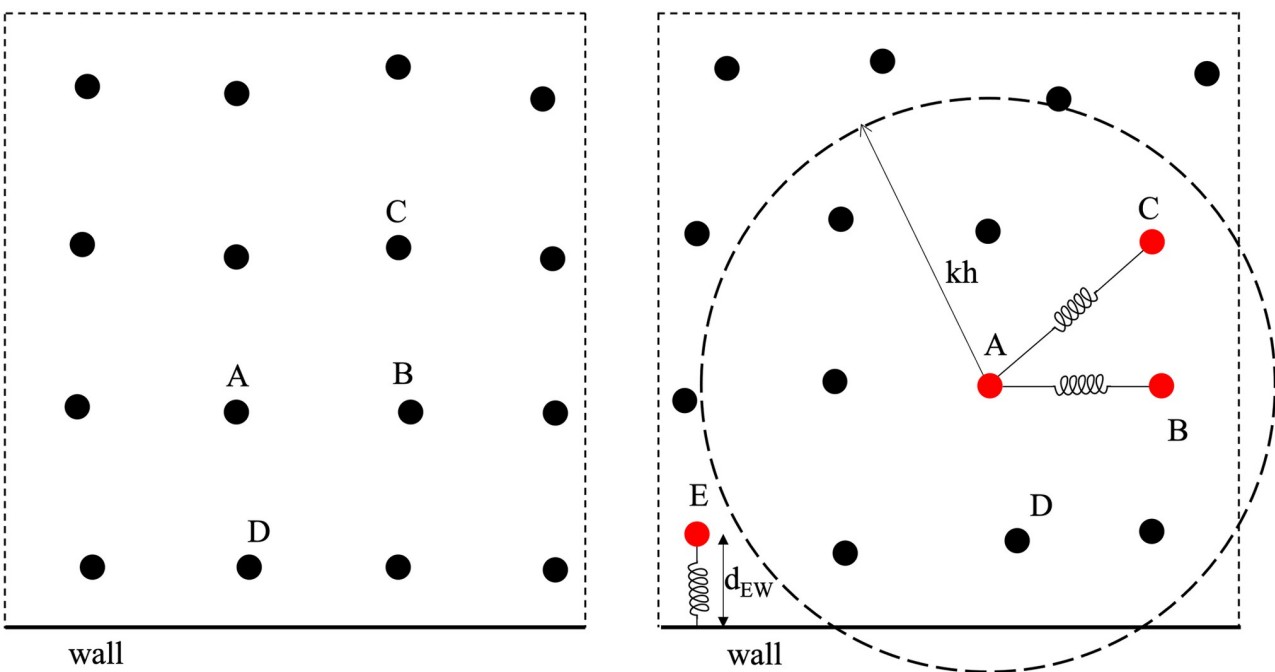

**Fig 4. Sketch of the solid particle formation.** Black full circles: fluid particles; red full circles: solid particle; bold black line: wall. a) Time instant r; b) time instant r+1.

described in section 2. At the r + 1 time step (see Fig 4b), the concentration of $C_{bp}$ for particles A, B, C and E exceeds the threshold value. These particles, represented in figure as red full circles, are converted into solid particles by introducing internal elastic forces to simulate the solid behavior. A procedure similar to that proposed by Monteleone et al. [55] is adopted here to obtain the elastic forces acting on the solid particles. However, instead of separating the fluid and solid domains through FSI-interfaces, in the proposed model, particles are simply switched from fluid to solid phase by adding spring links. This approach does not require the identification of the interface separating the two media. Specifically, when a fluid particle becomes a solid particle, it is linked to the neighboring solid particles having a distance less than kh from it (particles lying in the dashed circle). In Fig 4b, the particle A is linked to B and C through springs, whilst E is not connected to A, being the distance between the two particles, $d_{AE}$, greater than kh.

In principle, the links between solid particles were set not to change in time, so that the i solid particle maintains the same neighboring p solid particles. Each pair of mass points i-p is connected via a spring having an elastic constant $k_e$ and a rest length $l_{0,ip}$. In the model, the rest length of the spring is set equal to $l_{0,ip} = \Delta_x \sqrt{2}$, as this choice was verified to lead to better numerical stability in the solutions. In fact, this distance corresponds to the average distance between the particles in the reference starting configuration, and to the distance that the particles tend to reach in any generic distribution when thrombus starts to form. In order to reach and preserve the rest length, the springs respond by applying internal forces. Indicating with $l_{ip}$ the instantaneous updated distances from the i solid particle to the neighboring solid particles, the total internal force per unit mass acting on i, $\mathbf{f}_i$, can be expressed as

$$\boldsymbol{f}_i = \frac{k_e \Delta_x}{m_i} \sum_{p=1}^{N_p} \left( l_{0,ip} - l_{ip} \right) \hat{\boldsymbol{x}}_{ip} \tag{21}$$

where the summation is extended to the total number $N_p$ of solid particles connected to i and $\hat{\mathbf{x}}_{ip} = \left( \mathbf{x}_i - \mathbf{x}_p \right)/l_{ip}$ is the unit vector directed from i to p.

The force $\mathbf{f}_i$ is introduced in the momentum equation as body force per unit mass. Specifically, in the context of the ISPH approach, $\mathbf{f}_i$ is added in the predictor step Eq 2.

In order to handle the elastic deformation of the thrombus, a relationship between the spring constant ke and the structure mechanical properties was obtained using a procedure described by Monteleone et al. [55]. Specifically, a solid cube discretised with SPH particles bounded with springs was used to perform a tension stress analysis. In the test, several ke values were investigated and the associated Young's modulus E of the material was measured. As a result, a basic linear relationship between the spring coefficient and the Young's modulus was identified, where ke/E = 6.31.

Moreover, the solid particles having distance to the boundary less than $\Delta_x/2$ are linked to the wall through a spring to model the adhesion of the thrombus to the vessel. In Fig 4b, the solid particle E has a distance $d_{Ew} < \Delta_x/2$ and thus it is connected to the wall.

In this study, potential thrombus dissolution is allowed through a procedure similar to that proposed by Tosenberger et al. [56] to model platelets adhesion. In particular, once the spring forms, it can be stretched up to a threshold distance $l_{max}$ (corresponding to a limit local force), beyond which the link between the two particles is removed. Therefore, single particles or groups of particles bonded together can separate from the main thrombus and embolise. In Fig 5 the length of the spring connecting the particles A and C become larger that $l_{max}$ and thus the tie is removed. Since the spring connecting C and F is still active, and the clot

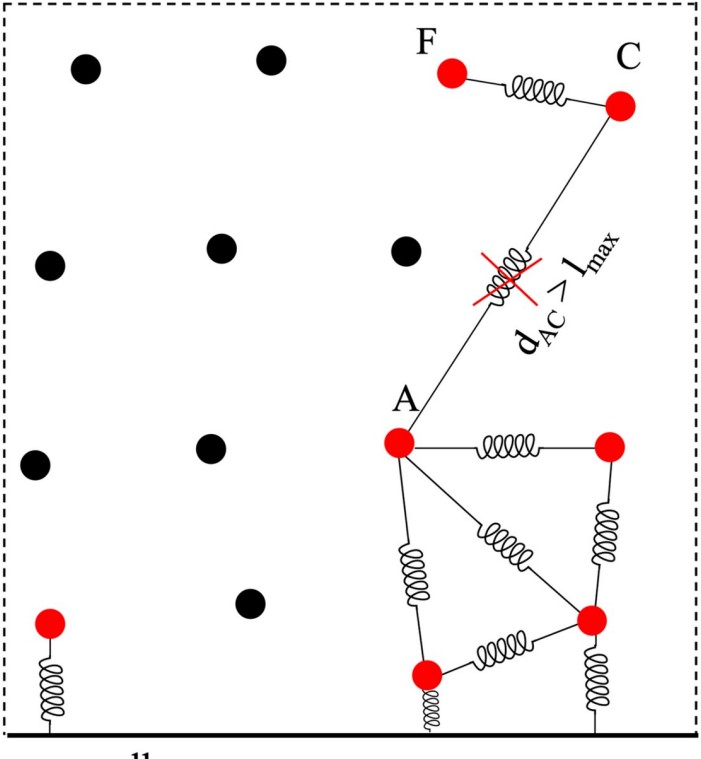

**Fig 5. Sketch of the solid particle separation.** Black full circles: fluid particles; red full circles: solid particle; bold black line: wall.

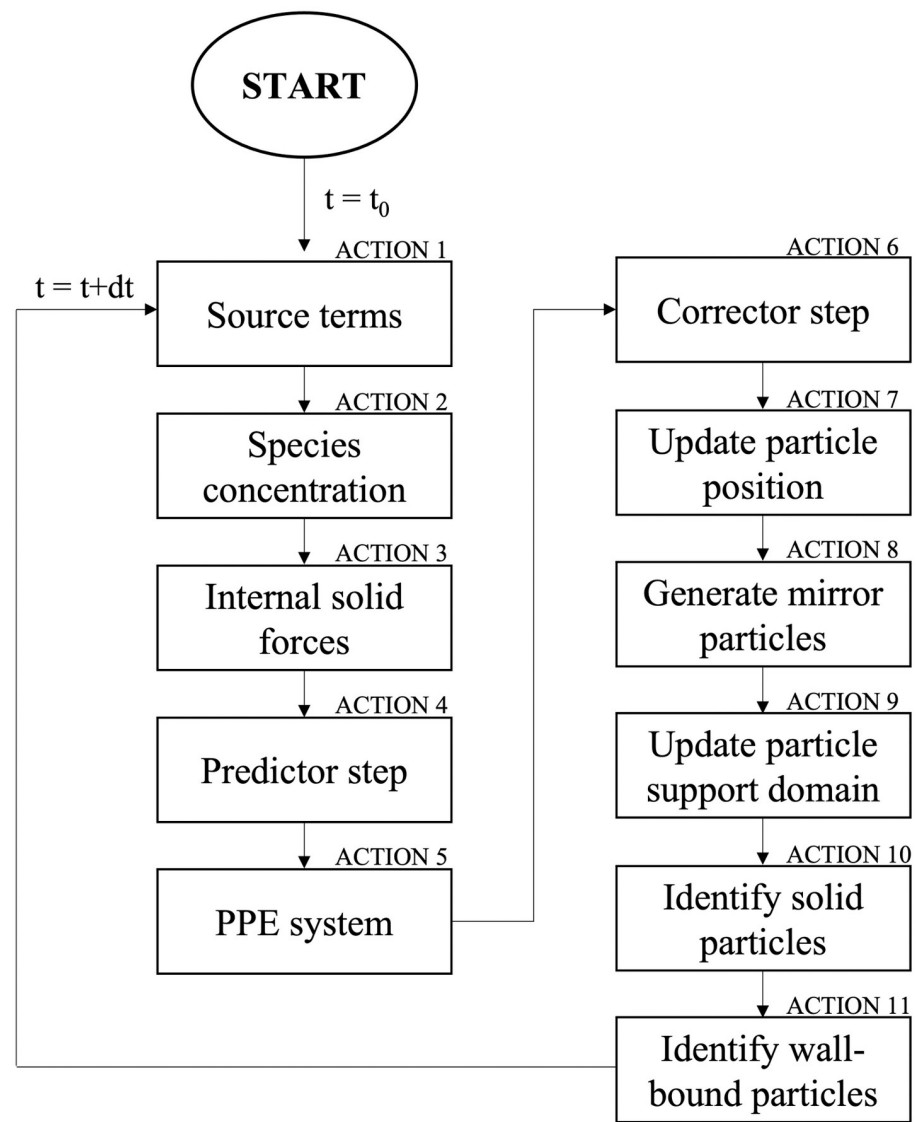

**Fig 6. Flow-chart of the proposed thrombus formation model.**

including particles C and F is now detached from the main thrombus and can be carried by the flow and expand recruiting other particles from the fluid.

A flow-chart providing a step by step description of the implemented thrombus formation model is reported in Fig 6. The actions indicated in the flow chart are briefly explained in the following:

- ACTION 1 –Source term: The source terms of the modelled species (pt, th, fg, fi, rp, ap and bp) to be introduced in the convection-diffusion equation are determined for each particle according to Eqs 9–15;

- ACTION 2 –Species concentration: The concentration of the modelled species is calculated for each particle through the convection-diffusion Eq 7;

- ACTION 3 –Internal solid forces: For each solid particle, the total force resulting from the system of neighbouring springs is calculated through Eq 21;

- ACTION 4 –Predictor step: In this step, Eq 2 is solved to calculate the intermediate velocity $\mathbf{u}^*$. For the solid particles, the force calculated at ACTION 3 is added to Eq 2;

- ACTION 5 –PPE system: The pseudo-pressure $\psi$ is calculated solving the system made up of one PPE (Eq 4) for each particle;

- ACTION 6 –Corrector step: In this step, the intermediate velocities are corrected obtaining the updated velocities (Eq 5);

- ACTION 7 –Update particle position: After calculating the updated velocity field, the particles are moved. The updated position $\mathbf{x}_i^{r+1}$ can be obtained using the mean value of the new and old velocities ($\mathbf{u}_i^{r+1}$ and $\mathbf{u}_i^r$, respectively);

- ACTION 8 –Generate mirror particles: The mirror particles are generated and the boundary conditions for the modelled species are imposed, as discussed in section 3.2;

- ACTION 9 –Update particle support domain: The support domain of each particle is determined including all the surrounding particles with distance lower than kh;

- ACTION 10—Identify solid particles: For each fluid particle, it is checked if the concentration of bound platelets exceeds the imposed threshold value. If this condition occurs, the particle is treated as solid and its list of neighbouring solid particles is created. Moreover, for each solid particle, the list of springs is updated during the simulation to bind new neighbours solid particles or to unbind particles that have exceeded the set distance threshold ($l_{max}$). The last condition is used to model potential thrombus dissolution.

- ACTION 11 –Identify wall-bound particles: The solid particles to be linked to the wall are identified. To this aim, the distance from the wall is determined and if it is less than $\Delta x/2$, a bond with the wall is introduced.

After ACTION 11, the simulation time is advanced by one time step (t = t+dt), and the procedure is reiterated from ACTION 1.

## 4. Results and discussion

### 4.1 Thrombus formation in backward facing step

To validate the proposed thrombus model, the case study proposed by Taylor *et al.* [41] was replicated, where the thrombus growth was analysed in a Backward-facing step (BFS). BFS configurations are often employed to analyse phenomena where the presence of flow separation and reattachment regions is a leading factor [57]. In fact, these geometries are easy to adapt to the different engineering problems, thanks to the direct control on the fluid domain by parameters such as the Reynolds number and the geometrical dimensions of the channel and step. The BFS geometry and dimensions selected by Taylor *et al.* [41] are represented in Fig 7.

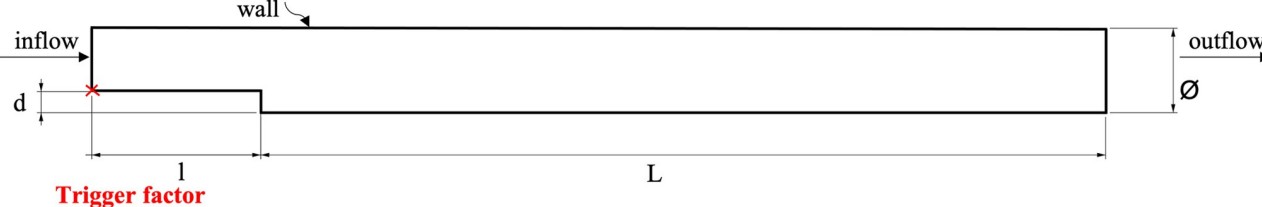

**Fig 7. Backward facing step scheme.** Boundary Conditions imposed; Geometry and dimension: s = 2.5 mm; Ø = 10 mm; l = 20 mm; L = 100 mm.

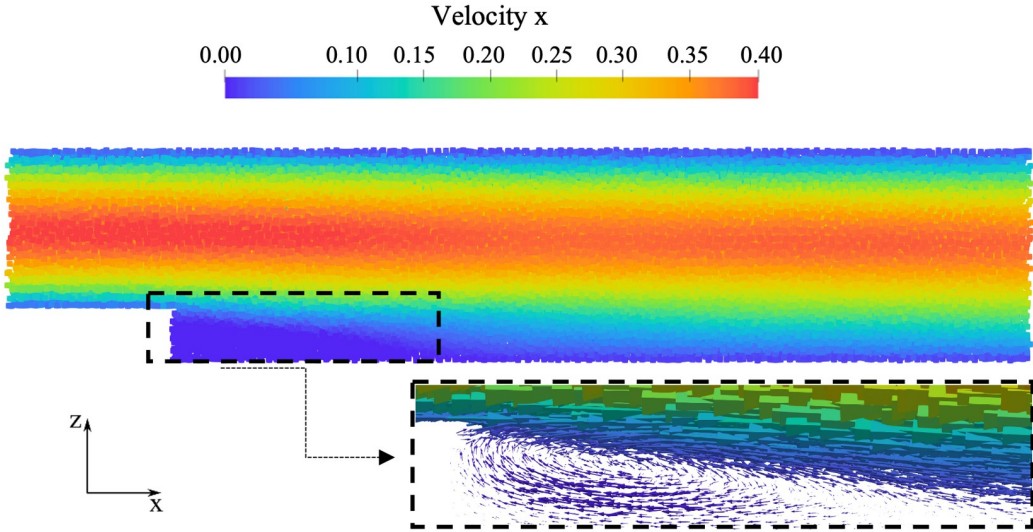

**Fig 8. Streamwise particle velocity [m/s] in BFS without thrombus.** An enlargement of the region near the step is highlighted with the velocity vectors.

In the simulation, the kernel width was set to 0.7 mm, resulting in a total of 215,000 particles. A flow rate of 0.76 l/min was imposed at the inlet section, whilst zero pressure was set at the outflow section (see Fig 7). In this study, the procedure described in Monteleone *et al.* [46] is used to handle open boundaries. This ensures mass conservation through the introduction of new particles in the computational domain, through the inflow section, balancing the particles which leave it through the outlet. Blood was modelled as a Newtonian fluid (the shear-thinning behaviour and yielding was neglected [58]), with density and dynamic viscosity equal to 1,060 kg/m$^3$ and 0.0035 Pa s, respectively. The resulting Reynolds' number was equal to 460, well within the laminar flow regimen.

In the numerical analysis, the parallel scheme of Monteleone *et al.* [59] was employed to save computational costs.

A preliminary hydrodynamic analysis was performed on the steady-state simulation without thrombus model (see Fig 8), to identify the recirculation region, obtaining a reattachment length equal to 17 mm, about 6.8 times the step height, consistently with Taylor *et al.* [41].

As previously described, the formation of blood clots is regulated by a complex cascade of biochemical reactions taking place on the surface of a growing thrombus. The transport of enzymes in the coagulation is reported to be dominated by convection phenomena in the bulk, whilst becomes diffusion-dominated in the zones characterised by low velocities [15]. Flow can both promote and limit thrombin generation and thrombus formation; and this behaviour makes its prediction a challenging task.

Recent investigations [60, 61] have indicated that thrombin generation is modified by variations in shear strain rate (SSR) and, in particular, its presence reduces where SSR is high. Due to the multi-scale nature of the problem, some strategies were used to speed up the computation. In the presented model, the SSR parameter was selected to amplify the different behaviour between bulk region and low velocity zones. Specifically, as described in section 3.2, since thrombin is selected as trigger factor, an amplification value equal to 10$^5$ was employed for the diffusive coefficient and source term of thrombin. Moreover, this coefficient was related to the SSR by applying it to particles having SSR lower that an imposed threshold value. Due to the

accelerated conversion of thrombin, a continuous supply of the inactive biochemicals (*pt* and *fg*) and resting platelets was imposed.

The concentrations of the modeled species were initialised in the bulk through a steady-state simulation imposing the typical values in healthy human blood (as reported in Table 1) except for the concentrations of thrombin, fibrin and bound platelet, that were set to zero. Moreover, an initial concentration of activated platelets was considered to simulate primary hemostasis. This value is equal to 5% of the background concentration of resting platelets, consistently with the recommendation from Sarrami-Foroushani *et al.* [4].

The biochemical reaction kinetic constants and the threshold value for the thrombin used in the simulation are reported in Table 1. A Young's modulus equal to 200 Pa (defined through the linear relationship between the Young's modulus and the spring constant $k_e$ described in section 3.3) was used in the analysis to simulate the early stage of the thrombus formation.

In this study, the threshold distance $l_{max}$ considered for the springs dissolution (as discussed in section 3.3) was set equal to 1.1 kh, as this value was found to lead to improved numerical stability and better correlation with experimental findings [41]. The trigger factor condition was imposed at the middle point of the step corner, as indicated in Fig 7, with a concentration of thrombin at the injured wall equal to the threshold value for thrombin amplified with the factor specified above ($C_{w,th} = 10^5 C_{th}^*$).

A qualitative analysis was performed to evaluate the best match value of the SSR threshold. Fig 9 illustrates a comparison in the shape taken by the simulated thrombus for four different values. As it can be observed, using a SSR threshold equal to 10 s$^{-1}$ (as recommended by Menichini and Xu [15]) the thrombus front is about triangular, with shape and dimensions that well resemble the experimental results described by Taylor *et al.* [41] (see Fig 10b).

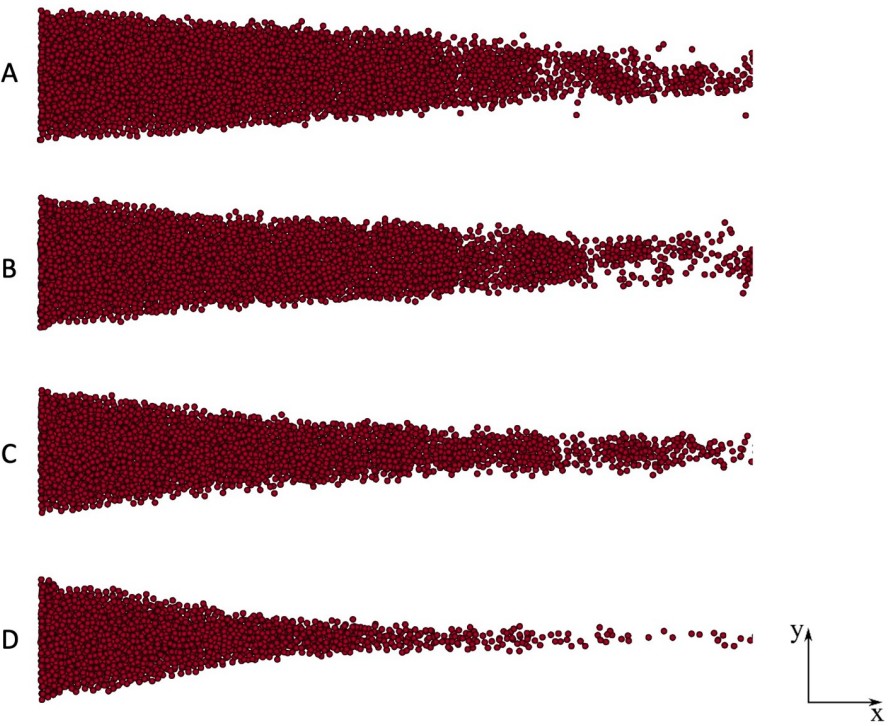

**Fig 9. Numerical results of thrombi formed (final time) considering various SSR threshold.** A) 40 s$^{-1}$; B) 30 s$^{-1}$; C) 20 s$^{-1}$; D) 10 s$^{-1}$.

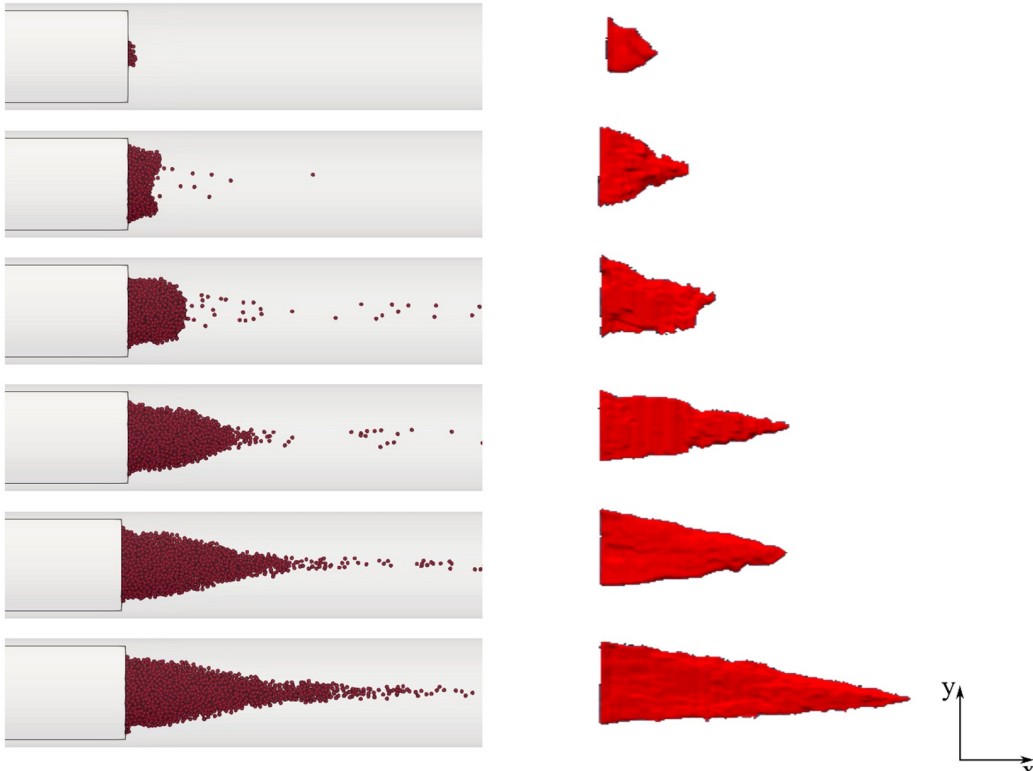

**Fig 10. Thrombus growth within the BFS.** a) Numerical results of FSI monolithic approach; b) Experimental results of [41] shown for comparison. Adapted from Taylor et al. [41] Copyright © 2014 ASME.

The temporal evolution of thrombus, in terms of formation and growth, is represented in Fig 10. The thrombus formation increases in both the radial and axial directions. Initially, the thrombus height reaches the step height (2.5 mm), while the maximum thrombus length achieved is equal to ten times the step height (25 mm). This is in agreement with the experimental results observed by Taylor *et al.* [41].

To assess the robustness and efficiency of the FSI monolithic method, the velocity field in the close proximity of the thrombus was examined. As shown in Fig 11, the thrombus growth is totally enclosed within the initial recirculation region, which eventually becomes entirely occupied by blood clotting.

The velocity profile at a cross-section positioned at an axial distal distance from the step equal to 8 mm, is plotted in Fig 12 at three time instants. The diagrams show the temporal decrease of velocity as the thrombus propagates and approaches the cross section, as described in Fig 11.

This confirms the mutual interaction between hydrodynamic variables and blood clotting. Furthermore, the observed thrombus length reaches an asymptotic value that relates to the reattachment length of the initial recirculation region. It should be noted that, exceeding this asymptotic length, as the clot increases downstream the step it is dragged due to high velocity field. Therefore, although particles reach the conditions at which they can become solid, they cannot bind to the main thrombus, which maintains its size. The thrombus lysis is simulated using only maximum distances (as discussed in section 3.3). Correlation with the maximum fibrin concentration might reproduce the process more realistically, and may be incorporated in the code in a future iteration.

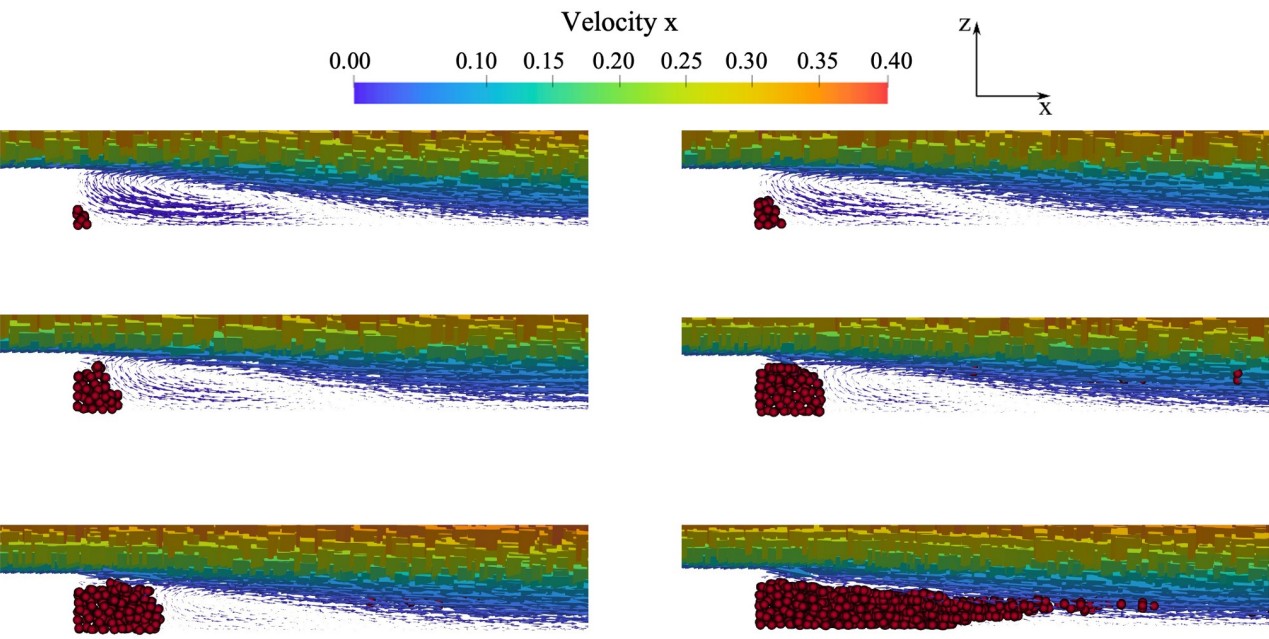

**Fig 11. Effect of thrombus formation on the velocity field in the vicinity of the step.**

An analysis of the species concentration is shown in Fig 13. Considering three time instants (indicated as $t_1$, $t_2$ and $t_3$ in the figure), the modelled active biochemical species (thrombin and fibrin) and platelets are triggered starting from the step position (see Fig 7) and spread downstream the channel.

The time evolution for the concentrations of the considered clotting factors were analysed at five different points in the recirculation region. These points are indicated in Fig 14. Points

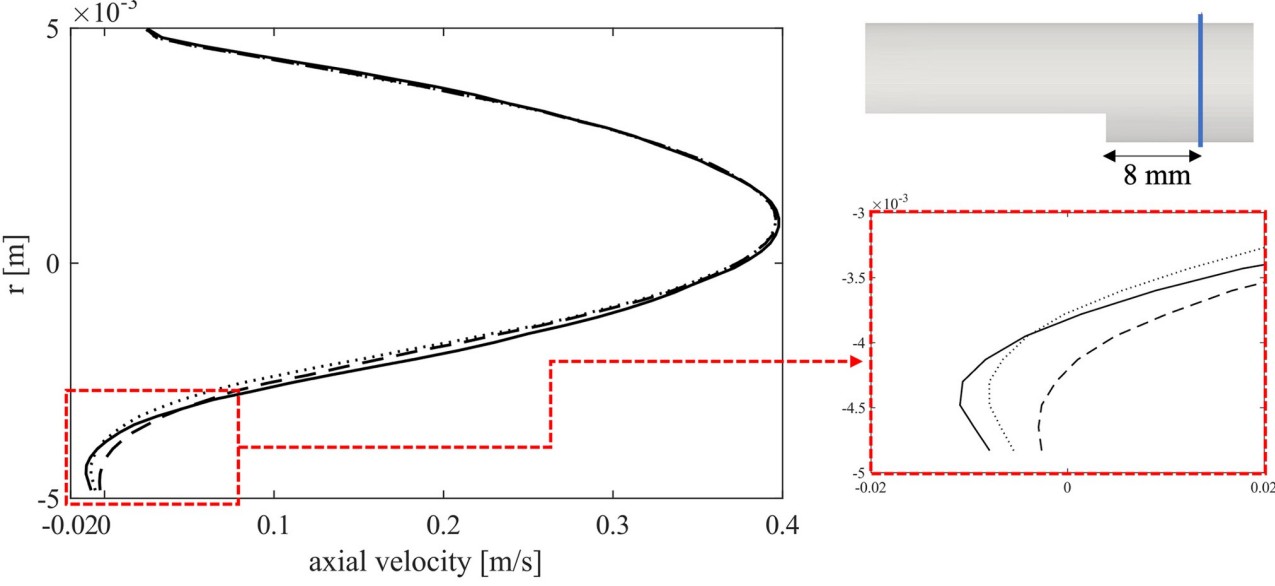

**Fig 12. Axial velocity profile at three time instants with a zoom of the region close to the step.** Continuous line: velocity at time zero; dotted line: time 0.5 s; dashed line: time 1 s.

**Fig 13. Biochemical specie concentration map (logarithmic scale) at three time instants: $t_1 = 0.5$ s; $t_2 = 1.5$ s; $t_3 = 3$ s.**

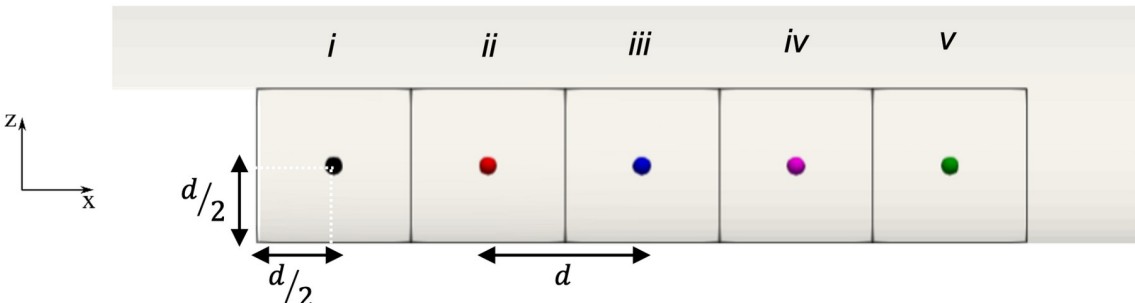

**Fig 14. Location of the measurement points used to evaluate the mean concentration of each specie.**

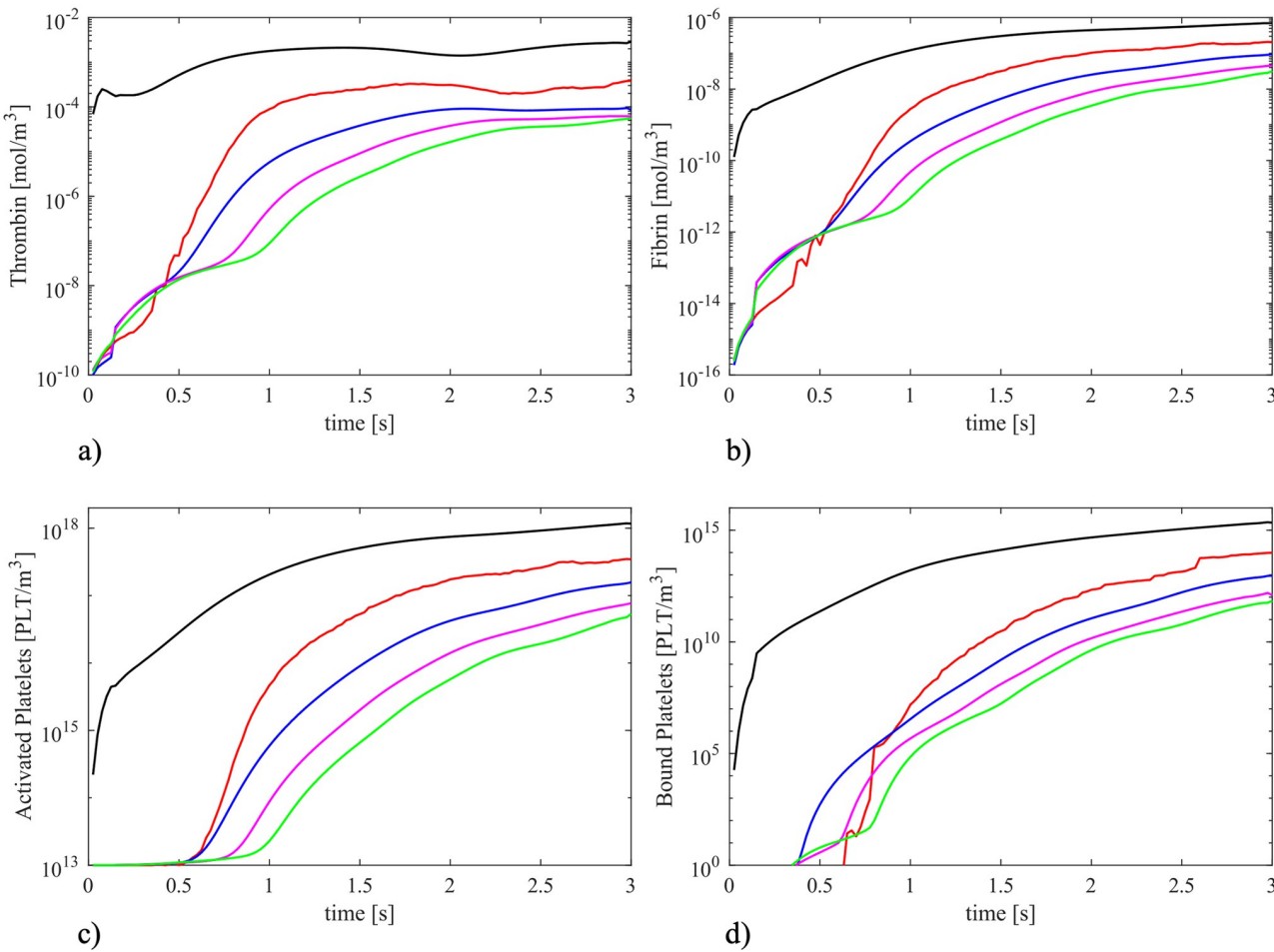

**Fig 15. Time evolution of the selected clotting factors considering five different points (see Fig 14).** Black continuous line: point A; red dotted line: point B; blue dashed line: point C; magenta dashed line: point D; green dashed dotted line: point E. Plotted used logarithmic scale in concentration axis.

are positioned in the plane of symmetry, evenly spaced with pace equal to $d$ (where $d$ is the step height), with the first point $i$ located at a distance equal to $-d/2$ from the step. The concentrations are calculated by averaging the values of particles lying at distance less than $kh$ from the considered central point. The graphs in Fig 15 show that all concentrations increase gradually until they reach an asymptotic value. Thrombin (Fig 15a) starts from a high value at point $i$ as a consequence of the trigger factor imposed at the step, whist the other points ($ii$ to $v$) have an initial null value. A similar trend is highlighted in Fig 15b for fibrin evolution. Activated platelets (Fig 15c) have an initial value equal to $10^{13}$ PLT/m$^3$, which steeply increases after about 0.5 s at points ii to v, as the activation function $\Omega$ becomes greater than 1 (see Eq 16). Since bound platelets are dependent on the activated platelets and fibrin mesh concentrations, their concentration follows the same development as represented in Fig 15d. The concentration analysis of the coagulative players of the thrombus formation process could be a helpful tool to individuate the regions involved by major risk stopping the main trigger factors.

## 5. Conclusions

The presented method is able to realistically describe the thrombosis phenomenon, including blood clotting variables (platelets, coagulative cascade enzymes), hydrodynamic parameters

(shear strain rate and velocity) and mutual interactions between fluid dynamics of blood and the forming thrombus.

Differently from the partitioned FSI approach, in the proposed technique, no interface is requested and an intrinsic coupling between fluid and solid phases is performed saving computational costs. Thanks to this feature, future developments will be addressed to encompass the effect of tissues deformation which are key aspects in the cardiovascular field. Moreover, the method here presented, exploiting the advantage of a Lagrangian meshfree particle method like SPH, could become very suitable to handle complex geometries of patient-specific models, such as left atrial appendages, heart valves and stenoses, where a complete understanding of the process is essential to predict the risk of thrombosis diseases. Furthermore, as a consequence of its simplicity and flexibility, the approach can be implemented into a new tool supporting the design of safer and more effective medical devices.

## Author Contributions

**Conceptualization:** Alessandra Monteleone, Alessia Viola, Enrico Napoli, Gaetano Burriesci.

**Formal analysis:** Alessandra Monteleone, Alessia Viola.

**Methodology:** Alessandra Monteleone, Alessia Viola, Gaetano Burriesci.

**Project administration:** Gaetano Burriesci.

**Software:** Alessandra Monteleone, Alessia Viola, Enrico Napoli.

**Supervision:** Alessandra Monteleone, Enrico Napoli, Gaetano Burriesci.

**Validation:** Alessandra Monteleone, Alessia Viola.

**Visualization:** Alessandra Monteleone, Alessia Viola.

**Writing – original draft:** Alessandra Monteleone, Alessia Viola.

**Writing – review & editing:** Enrico Napoli, Gaetano Burriesci.

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
