## [Decision Letter · Decision Letter 0]

1 Dec 2022

PONE-D-22-30268Modelling of thrombus formation using smoothed particle hydrodynamics methodPLOS ONE

Dear Dr. Burriesci,

Thank you for submitting your manuscript to PLOS ONE. After careful consideration, we feel that it has merit but does not fully meet PLOS ONE’s publication criteria as it currently stands. Therefore, we invite you to submit a revised version of the manuscript that addresses the points raised during the review process. Please revise your manuscript based on the Reviewers comments. I also invite the authors to highlight the novelty of their work, especially considering, as Reviewer 2 pointed out, that the authors were not aware of several articles published in the literature with subject similar to their manuscript.    

We look forward to receiving your revised manuscript.

Kind regards,

Alessio Alexiadis

Academic Editor

PLOS ONE

“The funders had no role in study design, data collection and analysis, decision to publish, or preparation of the manuscript”

“NO authors have competing interests”

Additional Editor Comments:

Please revise your manuscript based on the Reviewers comments

Reviewers' comments:

Reviewer's Responses to Questions

**Comments to the Author**

1. Is the manuscript technically sound, and do the data support the conclusions?

Reviewer #1: Yes

Reviewer #2: Yes

2. Has the statistical analysis been performed appropriately and rigorously? 

Reviewer #1: N/A

Reviewer #2: N/A

3. Have the authors made all data underlying the findings in their manuscript fully available?

Reviewer #1: No

Reviewer #2: No

4. Is the manuscript presented in an intelligible fashion and written in standard English?

Reviewer #1: Yes

Reviewer #2: Yes

5. Review Comments to the Author

Reviewer #1: A SPH-based FSI method, combined with a species transport equation, is proposed to simulate thrombus formation. Equation (21) is then used to form new bonding pair.

The main novelty is on the proposal of platelet activation/concentration model (Eq 6) for various species. The fluid particle is switched to solid particle if the concentration reaches a threshold value.

I have a few comments:

Comments:

1. A flow chart detailing on the FSI method and its coupling with the concentration model (6) should be given.

2. There are various constants tabulated in Table 1. Some references were cited for each constant. Are these constants generic and universal to all flow circumstances? Please discuss.

3. The threshold values for platelet activation, as well as methods on how to determine them, are not detailed.

4. Eq (21), the spring constant ke, is not given and discussed. How to determine this spring constant ke for a specific flow case?

5. I wonder why a factor of root(2) is needed for the rest length expression. I presume that root(2) is needed for diagonal neighbours, but not others (horizontal and vertical neighbors). Please explain.

6. The numerical and experimental observations are compared in Figure 8. Please explain how to tune the constants in the species model & other spring constants (if tuning is performed) during the validation stage. The C*th value is not given in this case.

Reviewer #2: General comments :

I reviewed the article written by Monteleone et Al. The work entitled Modelling of thrombus formation using smoothed particle hydrodynamics method presents a sph approach including an agglomeration algorithm to simulate the hemodynamics and the solid blood accrual in the flow.

The aim of the study is to show that, with a single model, is possible to simulate blood flow and thrombosis formation including coagulative cascade.

Major comments:

The bibliographical section is incomplete. Indeed, for instance some relevant papers exploring this research field are not presented, such as (non exhaustive list) :

• Modelling Particle Agglomeration on through Elastic Valves under Flow Baksamawi, HA; Ariane, M; (...); Alexiadis, A

• Using Discrete Multi-Physics for studying the dynamics of emboli in flexible venous valves Ariane, M; Vigolo, D; (...); Alexiadis, A

• Modeling Clot Formation of Shear-Injured Platelets in Flow by a Dissipative Particle Dynamics Method. Liwei Wang, Zengsheng Chen, Jiafeng Zhang, Xiwen Zhang & Zhongjun J. Wu

• A multiscale biomechanical model of platelets: Correlating with in-vitro results By:Zhang, P (Zhang, Peng) [1] ; Zhang, L (Zhang, Li) [2] ; Slepian, MJ (Slepian, Marvin J.) [1] , [3] , [4] ; Deng, YF (Deng, Yuefan) [2] ; Bluestein, D (Bluestein, Danny) [1]

• Multiscale Modeling of Mechanotransduction Processes in Flow-Induced Platelet Activation By:Gao, C (Gao, Chao) [1] ; Zhang, P (Zhang, Peng) [1] ; Bluestein, D (Bluestein, Danny) [1]

• Particle Method Simulation of Thrombus Formation in Fontan Route By:Tsubota, K (Tsubota, Ken-ichi) [1] ; Sughimoto, K (Sughimoto, Koichi) [2] ; Okauchi, K (Okauchi, Kazuki) [1] ; Liu, H (Liu, Hao) [1]

• Analysis of microvascular thrombus mechanobiology with a novel particle-based model By:Masalceva, AA (Masalceva, Anastasia A.) [1] , [2] ; Kaneva, VN (Kaneva, Valeriia N.) [1] , [2] , [3] ; Panteleev, MA (Panteleev, Mikhail A.) [1] , [2] , [3] , [4] ; Ataullakhanov, F (Ataullakhanov, Fazoil) [1] , [2] , [3] , [4] ; Volpert, V (Volpert, Vitaly) [5] , [6] , [7] ; Afanasyev, I (Afanasyev, Ilya) [8] , [9] ; Nechipurenko, DY (Nechipurenko, Dmitry Yu) [1] , [2] ,

• A Molecular Dynamics Based Multi-scale Platelet Aggregation Model and Its High-Throughput Simulation By:Xu, ZP (Xu, Zhipeng) [1] ; Zou, QS (Zou, Qingsong) [2]

• Particle hydrodynamic simulation of thrombus formation using velocity decay factor. Wang F, Xu S, Jiang D, Zhao B, Dong X, Zhou T, Luo X.

Minor comments:

• Is there any convergence criterion for this simulation especially for the agglomeration algorithm? Or is it an open-loop computation?

• What kind of particle does the author use at inlet section, periodic condition (if so, are the particle properties reset after the loop) or new particles.

• Section 3.3. references are needed to support the following sentence (‘the species do not affect the blood velocity in normal conditions’)

• Last phrase before section 4. Should be developed (‘the thrombus dissolution …….thrombus and embolise’)

• Figure 6: velocity vectors are too small, I suggest using a constant length instead and including colors for values gradient.

6. PLOS authors have the option to publish the peer review history of their article (what does this mean?). If published, this will include your full peer review and any attached files.

Reviewer #1: No

Reviewer #2: No

---

## [Author Response · Author response to Decision Letter 0]

11 Jan 2023

RESPONSE TO EDITOR

COMMENT: 

I also invite the authors to highlight the novelty of their work, especially considering, as Reviewer 2 pointed out, that the authors were not aware of several articles published in the literature with subject similar to their manuscript.

RESPONSE:

We are glad that you confirm that the paper is generally well written, and we appreciate your comment, and all competent and constructive feedback from yourself and the reviewers, which have been stimulating and we feel that has strongly contributed to improve the quality of the manuscript. All the issues mentioned by the reviewers have been addressed and a point-by-point reply is provided for each of them.

We have highlighted the novelty of our work in the revised manuscript:

“This paper presents a new three-dimensional numerical method based on SPH for the simulation of thrombus formation. Contrary to previous works, the proposed approach efficiently combines the biomechanical and biochemical processes in the thrombosis phenomenon. Four biochemical species and three platelets states are considered to replicate the main phases of the coagulative cascade. A particle agglomeration/dissolution algorithm is proposed, able to model both thrombus formation and growth, as well as embolisation. The fluid-solid coupling is enforced through to the inclusion of elastic forces between solid particles, which are established by recruiting fluid particles when specific hydrodynamic and biochemical conditions are satisfied. An innovative monolithic FSI approach is developed to describe the interaction between blood and the forming thrombus using a single solver.”

We trust that you and the reviewers will now find the manuscript acceptable for publication in the Plos One.

We look forward to hearing from you at your earliest convenience.

RESPONSE TO REVIEWER #1

We wish to express our gratitude to Referee 1 for their detailed and constructive comments, that have allowed us to largely improve the quality of the manuscript. In the new version of the manuscript, all points raised by Referee 1 are addressed and discussed. A point-by-point reply follows, where the changes to the original manuscript are explicitly listed.

COMMENT 1.1:

A flow chart detailing on the FSI method and its coupling with the concentration model (6) should be given.

RESPONSE 1.1:

We appreciate the reviewer’s suggestion. In the new version of the manuscript we have added a flow-chart of the proposed thrombus formation model. The new sentences that have been added in the revised manuscript are reported below:

“A flow-chart providing a step by step description of the implemented thrombus formation model is reported in Fig 6. The actions indicated in the flow chart are briefly explained in the following:

• ACTION 1 – Source term: The source terms of the modelled species (pt, th, fg, fi, rp, ap and bp) to be introduced in the convection-diffusion equation are determined for each particle according to eqns. 9-15;

• ACTION 2 – Species concentration: The concentration of the modelled species is calculated for each particle through the convection-diffusion eqn. 7;

• ACTION 3 – Internal solid forces: For each solid particle, the total force resulting from the system of neighbouring springs is calculated through eqn. 21;

• ACTION 4 – Predictor step: In this step, eqn. 2 is solved to calculate the intermediate velocity 𝒖∗. For the solid particles, the force calculated at ACTION 3 is added to eqn. 2;

• ACTION 5 – PPE system: The pseudo-pressure 𝜓 is calculated solving the system made up of one PPE (eqn. 4) for each particle;

• ACTION 6 – Corrector step: In this step, the intermediate velocities are corrected obtaining the updated velocities (eqn. 5);

• ACTION 7 – Update particle position: After calculating the updated velocity field, the particles are moved. The updated position xir+1can be obtained using the mean value of the new and old velocities (uir+1 and uir, respectively);

• ACTION 8 – Generate mirror particles: The mirror particles are generated and the boundary conditions for the modelled species are imposed, as discussed in section 3.2;

• ACTION 9 – Update particle support domain: The support domain of each particle is determined including all the surrounding particles with distance lower than kh;

• ACTION 10 - Identify solid particles: For each fluid particle, it is checked if the concentration of bound platelets exceeds the imposed threshold value. If this condition occurs, the particle is treated as solid and its list of neighbouring solid particles is created. Moreover, for each solid particle, the list of springs is updated during the simulation to bind new neighbours solid particles or to unbind particles that have exceeded the set distance threshold (lmax). The last condition is used to model potential thrombus dissolution.

• ACTION 11 – Identify wall-bound particles: The solid particles to be linked to the wall are identified. To this aim, the distance from the wall is determined and if it is less than Δ𝑥2⁄, a bond with the wall is introduced.

After ACTION 11, the simulation time is advanced by one time step (t = t+dt), and the procedure is reiterated from ACTION 1.”

COMMENT 1.2:

There are various constants tabulated in Table 1. Some references were cited for each constant. Are these constants generic and universal to all flow circumstances? Please discuss.

RESPONSE 1.2:

Constants used in the proposed approach and summarised in Table 1 of the manuscript are taken from the literature, and obtained from experimental tests on blood [4,12,52-55]. These constants have been used in a number of studies from other groups to model different flow conditions. For example, in [12,54] thrombus formation is simulated under steady flow conditions, while in [4] the clot in an aneurysm is modelled in pulsatile blood condition. We have included the following discussion in the revised manuscript:

“The constants and parameters used in the model are obtained from experimental studies available in the literature [4,12,52-55]. These values, listed in Table 1, are general and applicable in different flow conditions [4,12].”

COMMENT 1.3: 

The threshold values for platelet activation, as well as methods on how to determine them, are not detailed.

RESPONSE 1.3:

In the proposed model, the threshold value for the platelet activation is related to the thrombin concentration. In particular, we used the procedure proposed in [12], where the kinetic constant for the activated platelets kap is related to an activation function Ω through eqn. 16 of the manuscript.

In the expression of the activation function we assumed thrombin as the only agonist of the process (weigh 𝑤_𝑡ℎ = 1) with threshold concentration 𝐶*_𝑡ℎ = 9.11e-10 M and activation time t_act = 1, as recommended in [12,50].

Ω = 𝑤_𝑡ℎ 𝐶_𝑡ℎ / 𝐶*_𝑡ℎ.

In the proposed model, the conversion of resting platelets to activated platelets starts when the threshold concentration for thrombin is reached (𝐶_𝑡ℎ >= 𝐶*_𝑡ℎ) and thus the kinetic constant 𝑘_𝑎𝑝 in eqn. 16 of the manuscript becomes larger than 1.

We agree with the reviewer that including more details on the threshold value should be included in the manuscript. To cover this gap, the following description have been integrated in the section discussing the equation for the activation function:

“… where 𝑤_𝑡ℎ=1 and 𝐶*_𝑡ℎ is the threshold concentration for the thrombin. The latter was set equal to 9.11∙10^(-10) M, as recommended in the literature [12],[50] (see Table 1 of the manuscript). Therefore, conversion from resting to activated platelets is achieved when the concentration of the thrombin reaches the threshold value 𝐶*_𝑡ℎ”.

COMMENT 1.4:

Eq (21), the spring constant ke, is not given and discussed. How to determine this spring constant ke for a specific flow case?

RESPONSE 1.4:

The procedure to determine the spring constant is independent from the flow case. In particular, the relation between the spring coefficient k_e and the Young’s modulus E is determined following the technique implemented by group and detailed in [58]. Specifically, a cube of length L = 0.01 m was used to perform a tension stress analysis imposing a pressure of 10,000 Pa on the cube faces of normal direction z. All particles representing the cube were bounded with springs to simulate a solid behaviour. The Young’s modulus was thus obtained as the ratio of the imposed stress and the resulting strain in the z direction. The analysis was repeated to obtain a set of coefficients ke and the corresponding Young’s moduli. A linear relationship between the spring coefficient and the Young’s modulus was then identified (k_e/E = 6.31).

We agree with the reviewer that this point must be better clarified; therefore we have now included the following sentences in the revised manuscript:

“In order to handle the elastic deformation of the thrombus, a relationship between the spring constant ke and the structure mechanical properties was obtained using the procedure described by Monteleone et al. [58]. Specifically, a solid cube discretised with SPH particles bounded with springs was used to perform a tension stress analysis. In the test, several ke values were investigated and the associated Young’s modulus E of the material was measured. As a result, a basic linear relationship between the spring coefficient and the Young’s modulus was identified, where k_e/E = 6.31.”

COMMENT 1.5: 

I wonder why a factor of root(2) is needed for the rest length expression. I presume that root(2) is needed for diagonal neighbours, but not others (horizontal and vertical neighbors). Please explain.

RESPONSE 1.5:

We agree with the reviewer that the use of the square root for the rest length expression must be better explained. In the model, we use an average distance for the rest length expression. This was found to give solutions with better numerical stability. In particular, in the starting configuration the particles are arranged at an isotropic initial distance Δ𝑥, that in this study was assumed equal to the smoothing length h. In this configuration, a generic i particle, has a total of 26 particles in its support domain: 6 particles with distance Δ𝑥, 12 particles with distance Δ𝑥 sqrt(2) and 8 particles with distance Δ𝑥 sqrt(3). The mean distance is thus equal to Δ𝑥 sqrt(2). In a general configuration, such as that present at the thrombus formation, the particles are not distributed anymore in a regular way as in the reference configuration, but it is observed that they globally tend to reach this average distance.

In order to clarify that point, we have added the following sentences in the revised manuscript:

“In the model, the rest length of the spring is set equal to 𝑙_0, 𝑖𝑝 = Δ𝑥 √2, as this choice was verified to lead to better numerical stability in the solutions. In fact, this distance corresponds to the average distance between the particles in the reference starting configuration, and to the distance that the particles tend to reach in any generic distribution when thrombus starts to form.”

COMMENT 1.6:

The numerical and experimental observations are compared in Figure 8. Please explain how to tune the constants in the species model & other spring constants (if tuning is performed) during the validation stage. The 𝐶*_𝑡ℎ value is not given in this case.

RESPONSE 1.6:

The constant species and the 𝐶*_𝑡ℎ value used in the validation test are reported in Table 1 of the manuscript. These values are constant, and no tuning procedure was performed to calibrate them. Similarly, as explained in response to COMMENT 1.4 of this document, the spring constant k_e is fixed and related to the Young’s modulus E through the relation k_e/E = 6.31. In particular, a Young’s modulus of 200 Pa was used in the simulation to model the early stage of the thrombus formation. We agree with the reviewer that this needs to be better clarified in the manuscript, where we have added the following sentences:

“The biochemical reaction kinetic constants and the threshold value for the thrombin used in the simulation are summarised in Table 1. A Young’s modulus equal to 200 Pa (defined through the linear relationship between the Young’s modulus and the spring constant ke described in section 3.3) was used in the analysis to simulate the early stage of the thrombus formation.”

REFERENCES

[4] Sarrami-Foroushani A, Lassila T, Hejazi SM, Nagaraja S, Bacon A, Frangi AF. A computational model for prediction of clot platelet content in flow-diverted intracranial aneurysms. J Biomech. 2019;91: 7–13. doi:10.1016/j.jbiomech.2019.04.045

[12] Sorensen EN, Burgreen GW, Wagner WR, Antaki JF. Computational Simulation of Platelet Deposition and Activation: I. Model Development and Properties. Ann Biomed Eng. 1999;27: 436–448. doi:10.1114/1.200

[50] Weiss HJ. Platelets: Pathophysiology and Antiplatelet Drug Therapy. Liss. 1982. 

[52] Grunkemeier JM, Tsai WB, Horbett TA. Hemocompatibility of treated polystyrene substrates: Contact activation, platelet adhesion, and procoagulant activity of adherent platelets. J Biomed Mater Res. 1998;41: 657–670. doi:10.1002/(SICI)1097-4636(19980915)41:4<657::AID-JBM18>3.0.CO;2-B

[53] Rosing J, van Rijn J, Bevers E, van Dieijen G, Comfurius P, Zwaal R. The role of activated human platelets in prothrombin and factor X activation. Blood. 1985;65: 319–332. doi:10.1182/blood.V65.2.319.319

[54] Anand M, Rajagopal K, Rajagopal KR. A Model Incorporating Some of the Mechanical and Biochemical Factors Underlying Clot Formation and Dissolution in Flowing Blood. Journal of Theoretical Medicine. 2003;5: 183–218. doi:10.1080/10273660412331317415

[55] Tsiang M, Paborsky LR, Li W-X, Jain AK, Mao CT, Dunn KE, et al. Protein Engineering Thrombin for Optimal Specificity and Potency of Anticoagulant Activity in Vivo. Biochemistry. 1996;35: 16449–16457. doi:10.1021/bi9616108

[58] Monteleone A, Borino G, Napoli E, Burriesci G. Fluid–structure interaction approach with smoothed particle hydrodynamics and particle–spring systems. Comput Methods Appl Mech Eng. 2022;392: 114728. doi:https://doi.org/10.1016/j.cma.2022.114728

RESPONSE TO REVIEWER #2

We wish to express our gratitude to Referee 2 for his/her detailed and constructive comments, that have allowed us to largely improve the quality of the manuscript. In the new version of the manuscript, all points raised by Referee 2 are addressed and discussed. A point-by-point reply follows, where the changes to the original manuscript are explicitly listed.

MAJOR COMMENTS:

The bibliographical section is incomplete. Indeed, for instance some relevant papers exploring this research field are not presented, such as (non exhaustive list) :

• Modelling Particle Agglomeration on through Elastic Valves under Flow Baksamawi, HA; Ariane, M; (...); Alexiadis, A

• Using Discrete Multi-Physics for studying the dynamics of emboli in flexible venous valves Ariane, M; Vigolo, D; (...); Alexiadis, A

• Modeling Clot Formation of Shear-Injured Platelets in Flow by a Dissipative Particle Dynamics Method. Liwei Wang, Zengsheng Chen, Jiafeng Zhang, Xiwen Zhang & Zhongjun J. Wu

• A multiscale biomechanical model of platelets: Correlating with in-vitro results By:Zhang, P (Zhang, Peng) [1] ; Zhang, L (Zhang, Li) [2] ; Slepian, MJ (Slepian, Marvin J.) [1] , [3] , [4] ; Deng, YF (Deng, Yuefan) [2] ; Bluestein, D (Bluestein, Danny) [1]

• Multiscale Modeling of Mechanotransduction Processes in Flow-Induced Platelet Activation By:Gao, C (Gao, Chao) [1] ; Zhang, P (Zhang, Peng) [1] ; Bluestein, D (Bluestein, Danny) [1]

• Particle Method Simulation of Thrombus Formation in Fontan Route By:Tsubota, K (Tsubota, Ken-ichi) [1] ; Sughimoto, K (Sughimoto, Koichi) [2] ; Okauchi, K (Okauchi, Kazuki) [1] ; Liu, H (Liu, Hao) [1]

• Analysis of microvascular thrombus mechanobiology with a novel particle-based model By:Masalceva, AA (Masalceva, Anastasia A.) [1] , [2] ; Kaneva, VN (Kaneva, Valeriia N.) [1] , [2] , [3] ; Panteleev, MA (Panteleev, Mikhail A.) [1] , [2] , [3] , [4] ; Ataullakhanov, F (Ataullakhanov, Fazoil) [1] , [2] , [3] , [4] ; Volpert, V (Volpert, Vitaly) [5] , [6] , [7] ; Afanasyev, I (Afanasyev, Ilya) [8] , [9] ; Nechipurenko, DY (Nechipurenko, Dmitry Yu) [1] , [2] ,

• A Molecular Dynamics Based Multi-scale Platelet Aggregation Model and Its High-Throughput Simulation By:Xu, ZP (Xu, Zhipeng) [1] ; Zou, QS (Zou, Qingsong) [2]

• Particle hydrodynamic simulation of thrombus formation using velocity decay factor. Wang F, Xu S, Jiang D, Zhao B, Dong X, Zhou T, Luo X.

RESPONSE TO MAJOR COMMENTS:

We thank the reviewer for the recommendation to expand the literature review presented in the manuscript, and for suggesting several relevant papers. The references advised by the reviewer have been added in the revised manuscript:

“Zhang et al. [8] and Gao et al. [9] implemented a novel multiscale approach based on discrete particle methods to model thrombus formation in cardiovascular diseases by coupling the macroscopic flow conditions with cellular and molecular effects of platelet mechanical activation. Xu et al. [10] proposed a multi-scale approach where fluid was simulated on the macro-scale using dissipative particle dynamics, and the fine-scale receptors’ biochemical reactions were modelled by coarse-grained molecular dynamics.

…

Recently, a number of particle techniques had been developed to describe thrombosis. Tsubota et al. [30] presented a semi-implicit two-dimensional moving particle approach to model thrombus formation after Fontan surgery. In this model, fluid particles are converted into solid phase by adding internal spring forces when blood stasis condition occurs (the model does not consider biochemical factors). Masalceva et al. [31] developed a two-dimensional particle-based model including thrombus shell as aggregate of particles and thrombin specie. Also this approach neglects the biochemical reactions of the coagulation cascade and fibrin formation. Wang et al. [32] proposed a novel particle method to simulate thrombus formation employing a velocity decay factor linked to the fibrin concentration, to take into account the interaction with blood. Wang et al. [33] developed a dissipative particle dynamics model to study the adhesion and aggregation process of injured platelets on the collagen surface, by incorporating a model of high non-physiological shear stresses traumatised platelets to a viscoelastic model.

…

Ariane et al. [38] proposed a two-dimensional model to simulate the interaction between blood flow and emboli-like structures in a double venous valve system. In this approach no particle agglomeration is used to model emboli structures, that are considered as fixed. This technique was extended by Baksamawi et al. [39] including an algorithm based on geometrical distance for particle agglomeration.”

MINOR COMMENTS:

COMMENT 2.1: 

Is there any convergence criterion for this simulation especially for the agglomeration algorithm? Or is it an open-loop computation?

RESPONSE 2.1:

No convergence criterion was needed for this simulation. However, we will consider the opportunity to include one for future developments of the model.

COMMENT 2.2: 

What kind of particle does the author use at inlet section, periodic condition (if so, are the particle properties reset after the loop) or new particles.

RESPONSE 2.2:

New particles are introduced at the inlet section following the procedure described in Monteleone et al. [46]. This technique allows to handle open boundaries, guarantying a correct mass conservation through the balance of new particles which are continuously introduced in the computational domain through the inflow section and other particles which leave it through the outlet.

Following the reviewer’s suggestion, the following sentence has been included in the revised manuscript:

“In this study, the procedure described in Monteleone et al. [46] is used to handle open boundaries. This ensures mass conservation through the introduction of new particles in the computational domain, through the inflow section, balancing the particles which leave it through the outlet.”

COMMENT 2.3: 

Section 3.3. references are needed to support the following sentence (‘the species do not affect the blood velocity in normal conditions’)

RESPONSE 2.3:

We apologise about the misleading phrasing of the statement ‘the species do not affect the blood velocity in normal conditions’, which is an assumption based on our understanding of the phenomenon, rather than on specific studies in the literature which we have not been able to source.

In the revised manuscript, we have clarified that this is an assumption in the model, with the sentence modified as follows:

“Moreover, it was assumed that the concentrations of the species do not affect the blood velocity in normal conditions, in the absence of thrombus.”

COMMENT 2.4: 

Last phrase before section 4. Should be developed (‘the thrombus dissolution …….thrombus and embolise’)

RESPONSE 2.4:

We agree with the reviewer that the algorithm used to handle the thrombus dissolution should be discussed in detail. To this aim, one more figure has been added (Fig. 5 of the revised manuscript) and the manuscript has been modified as follows:

“In this study, potential thrombus dissolution is allowed through a procedure similar to that proposed by Tosenberger et al. [59] to model platelets adhesion. In particular, once the spring forms, it can be stretched up to a threshold distance lmax (corresponding to a limit local force), beyond which the link between the two particles is removed. Therefore, single particles or groups of particles bonded together can separate from the main thrombus and embolise. In Fig. 5 the length of the spring connecting the particles A and C become larger than lmax and thus the tie is removed. Since the spring connecting C and F is still active, and the clot including particles C and F is now detached from the main thrombus and can be carried by the flow and expand recruiting other particles from the fluid.”

Moreover, in section 4.1 (Thrombus formation in backward facing step) when discussing the validation test, the following sentences have been added in the revised manuscript:

“In this study, the threshold distance l_max considered for the springs dissolution (as discussed in section 3.3) was set equal to 1.1 k_h, as this value was found to lead to improved numerical stability and better correlation with experimental findings [41].”

COMMENT 2.5: 

Figure 6: velocity vectors are too small, I suggest using a constant length instead and including colors for values gradient.

RESPONSE 2.5:

We appreciate the reviewer suggestion. We have tried to modify the figure using vectors with constant length coloured by velocity values. However, we feel that this does not improve the clarity and be misleading, requiring the use of two different scales for the velocity. Hence, we have preferred to maintain the current representation.

REFERENCES

[8] Zhang P, Zhang L, Slepian MJ, Deng Y, Bluestein D. A multiscale biomechanical model of platelets: Correlating with in-vitro results. J Biomech. 2017;50: 26–33. doi:10.1016/j.jbiomech.2016.11.019

[9] Gao C, Zhang P, Bluestein D. Multiscale Modeling of Mechanotransduction Processes in Flow-Induced Platelet Activation. 2016 IEEE 2nd International Conference on Big Data Security on Cloud (BigDataSecurity), IEEE International Conference on High Performance and Smart Computing (HPSC), and IEEE International Conference on Intelligent Data and Security (IDS). IEEE; 2016. pp. 274–279. doi:10.1109/BigDataSecurity-HPSC-IDS.2016.13

[10] Xu Z, Zou Q. A Molecular Dynamics Based Multi-scale Platelet Aggregation Model and Its High-Throughput Simulation. 2022. pp. 81–92. doi:10.1007/978-3-030-96772-7_8

[30] Tsubota K, Sughimoto K, Okauchi K, Liu H. Particle Method Simulation of Thrombus Formation in Fontan Route. 2016. pp. 387–396. doi:10.1007/978-3-319-40827-9_30

[31] Masalceva AA, Kaneva VN, Panteleev MA, Ataullakhanov F, Volpert V, Afanasyev I, et al. Analysis of microvascular thrombus mechanobiology with a novel particle-based model. J Biomech. 2022;130: 110801. doi:10.1016/j.jbiomech.2021.110801

[32] Wang F, Xu S, Jiang D, Zhao B, Dong X, Zhou T, et al. Particle hydrodynamic simulation of thrombus formation using velocity decay factor. Comput Methods Programs Biomed. 2021;207: 106173. doi:10.1016/j.cmpb.2021.106173

[33] Wang L, Chen Z, Zhang J, Zhang X, Wu ZJ. Modeling Clot Formation of Shear-Injured Platelets in Flow by a Dissipative Particle Dynamics Method. Bull Math Biol. 2020;82: 83. doi:10.1007/s11538-020-00760-9

[38] Ariane M, Vigolo D, Brill A, Nash FGB, Barigou M, Alexiadis A. Using Discrete Multi-Physics for studying the dynamics of emboli in flexible venous valves. Comput Fluids. 2018;166: 57–63. doi:10.1016/j.compfluid.2018.01.037

[39] Baksamawi HA, Ariane M, Brill A, Vigolo D, Alexiadis A. Modelling Particle Agglomeration on through Elastic Valves under Flow. ChemEngineering. 2021;5: 40. doi:10.3390/chemengineering5030040

[41] Taylor JO, Witmer KP, Neuberger T, Craven BA, Meyer RS, Deutsch S, et al. In Vitro Quantification of Time Dependent Thrombus Size Using Magnetic Resonance Imaging and Computational Simulations of Thrombus Surface Shear Stresses. J Biomech Eng. 2014;136. doi:10.1115/1.4027613

[46] Monteleone A, Monteforte M, Napoli E. Inflow/outflow pressure boundary conditions for smoothed particle hydrodynamics simulations of incompressible flows. Comput Fluids. 2017;159. doi:10.1016/j.compfluid.2017.09.011

[59] Tosenberger A, Ataullakhanov F, Bessonov N, Panteleev M, Tokarev A, Volpert V. Modelling of platelet–fibrin clot formation in flow with a DPD–PDE method. J Math Biol. 2016;72: 649–681. doi:10.1007/s00285-015-0891-2

---

## [Decision Letter · Decision Letter 1]

24 Jan 2023

Modelling of thrombus formation using smoothed particle hydrodynamics method

PONE-D-22-30268R1

Dear Dr. Burriesci,

We’re pleased to inform you that your manuscript has been judged scientifically suitable for publication and will be formally accepted for publication once it meets all outstanding technical requirements.

Kind regards,

Alessio Alexiadis

Academic Editor

PLOS ONE

Additional Editor Comments (optional):

Reviewers' comments:

Reviewer's Responses to Questions

**Comments to the Author**

1. If the authors have adequately addressed your comments raised in a previous round of review and you feel that this manuscript is now acceptable for publication, you may indicate that here to bypass the “Comments to the Author” section, enter your conflict of interest statement in the “Confidential to Editor” section, and submit your "Accept" recommendation.

Reviewer #1: All comments have been addressed

Reviewer #2: All comments have been addressed

2. Is the manuscript technically sound, and do the data support the conclusions?

Reviewer #1: Yes

Reviewer #2: (No Response)

3. Has the statistical analysis been performed appropriately and rigorously? 

Reviewer #1: N/A

Reviewer #2: (No Response)

4. Have the authors made all data underlying the findings in their manuscript fully available?

Reviewer #1: No

Reviewer #2: (No Response)

5. Is the manuscript presented in an intelligible fashion and written in standard English?

Reviewer #1: Yes

Reviewer #2: (No Response)

6. Review Comments to the Author

Reviewer #1: The authors have addressed my technical points and comments made previously.

The paper can be accepted for publication.

Reviewer #2: (No Response)

7. PLOS authors have the option to publish the peer review history of their article (what does this mean?). If published, this will include your full peer review and any attached files.

Reviewer #1: No

Reviewer #2: No

---

## [Editor Report · Acceptance letter]

26 Jan 2023

PONE-D-22-30268R1 

Modelling of thrombus formation using smoothed particle hydrodynamics method 

Dear Dr. Burriesci:

I'm pleased to inform you that your manuscript has been deemed suitable for publication in PLOS ONE. Congratulations! Your manuscript is now with our production department. 

Kind regards, 

on behalf of

Dr. Alessio Alexiadis 

Academic Editor

PLOS ONE